# Conformational changes in the essential *E. coli* septal cell wall synthesis complex suggest an activation mechanism

Brooke M. Britton[1,3], Remy A. Yovanno [1,3], Sara F. Costa[2], Joshua McCausland[1], Albert Y. Lau [1], Jie Xiao [1] ✉ & Zach Hensel [2] ✉

The bacterial divisome is a macromolecular machine composed of more than 30 proteins that controls cell wall constriction during division. Here, we present a model of the structure and dynamics of the core complex of the *E. coli* divisome, supported by a combination of structure prediction, molecular dynamics simulation, single-molecule imaging, and mutagenesis. We focus on the septal cell wall synthase complex formed by FtsW and FtsI, and its regulators FtsQ, FtsL, FtsB, and FtsN. The results indicate extensive interactions in four regions in the periplasmic domains of the complex. FtsQ, FtsL, and FtsB support FtsI in an extended conformation, with the FtsI transpeptidase domain lifted away from the membrane through interactions among the C-terminal domains. FtsN binds between FtsI and FtsL in a region rich in residues with superfission (activating) and dominant negative (inhibitory) mutations. Mutagenesis experiments and simulations suggest that the essential domain of FtsN links FtsI and FtsL together, potentially modulating interactions between the anchor-loop of FtsI and the putative catalytic cavity of FtsW, thus suggesting a mechanism of how FtsN activates the cell wall synthesis activities of FtsW and FtsI.

Bacterial cell division is an essential process and a frequent target for novel antibiotics[1–3]. The cytoskeletal protein FtsZ defines the future division site[4] and recruits more than thirty proteins, many of them cell-wall enzymes and regulators. Collectively termed the divisome[5,6], these proteins orchestrate cell-wall constriction during cell division. In *Escherichia coli*, the essential septal peptidoglycan (sPG) polymerase FtsW and its cognate transpeptidase FtsI form the sPG synthase complex FtsWI and cooperate to synthesize new septal cell wall[7,8]. FtsWI activities are regulated by a conserved subcomplex of transmembrane proteins FtsQ, FtsL, and FtsB (FtsQLB)[9–12], and by FtsN in γ-proteobacteria[13]. However, how the septal cell wall synthesis activities of FtsWI are regulated by FtsQLB and FtsN remains unclear.

In *E. coli*, FtsN has been regarded as the trigger of septum synthesis, as its arrival at the division site coincides with the initiation of cell-wall constriction[13]. FtsN and FtsN-like proteins are conserved in γ-proteobacteria and contain SPOR domains that bind denuded glycans[14]. In *E. coli*, an essential, periplasmic segment of FtsN (FtsN[E]) is sufficient to initiate constriction in the absence of full-length FtsN[13,15,16]. Our recent single-molecule studies show that FtsN[E] is part of a processive sPG synthesis complex with active FtsWI and required to maintain the processivity of the FtsWI complex[17,18]. Whether FtsQLB is also part of this complex and, if so, how it associates with FtsWI, is unknown. In vitro experiments using purified proteins found that *E. coli* FtsQLB inhibits FtsI activity[19], while *Pseudomonas aeruginosa* FtsQLB enhances FtsW activity[9]. In neither case did the addition of FtsN impact FtsI or FtsW activity. In vivo experiments showed that mutations in FtsL, FtsW, and FtsI can be either dominant negative (DN) or superfission (SF, bypassing FtsN and/or complementing DN

[1]Department of Biophysics and Biophysical Chemistry, Johns Hopkins School of Medicine, 725 N. Wolfe St, Baltimore, MD 21205, USA. [2]ITQB NOVA, Universidade NOVA de Lisboa, Lisbon, Av. da República, 2780-157 Oeiras, Portugal. [3]These authors contributed equally: Brooke M. Britton, Remy A. Yovanno. ✉e-mail: xiao@jhmi.edu; zach.hensel@itqb.unl.pt

mutations)[9,10,15,20,21]. These seemingly contradictory observations suggest that FtsWI can transition between on and off states depending on the environment and/or genetic mutations, and that FtsQLB and FtsN may play important roles in shifting FtsWI between the two states[10,11,20]. However, it is unclear what conformations of FtsWI correspond to the on and off states and how FtsN shifts FtsWI between the two states.

A recent review synthesized evidence to date and used structure predictions to propose a model of allosteric regulation of FtsWI in which multiple interactions cooperatively stabilize an active conformation[22]. This work drew on structure predictions of complexes FtsWI, FtsQLB, and FtsLWI using AlphaFold2 (AF2)[23]. Structure prediction was also recently applied to other divisome interactions[20,24]. While AF2 makes predictions that are largely consistent with available structural data and can accurately predict protein–protein interfaces[25–27], its predictions can fail to distinguish between different states of the same complex[28] or describe the consequences of point mutations[29].

In this work, we provided experimental evidence to show that FtsQLB forms a complex with active FtsWI in vivo and conducted structure predictions for *E. coli* FtsWI, FtsWI in complex with FtsQLB (FtsQLBWI), and for FtsQLBWI in complex with FtsN^E (FtsQLBWIN). We subjected these predicted structures to all-atom molecular dynamics (MD) simulation to observe dynamics of protein–protein interfaces on the microsecond timescale. We observed extensive interactions in the transmembrane and periplasmic domains of the FtsQLBWI and FtsQLBWIN complexes. Next, we carried out mutagenesis and single-molecule tracking experiments to investigate these observed interactions. Further MD simulations using DN and SF point mutations revealed critical conformational changes at FtsQLB-FtsI protein–protein interfaces and near the putative catalytic region of FtsW. Collectively, our results support a model in which FtsQLB scaffolds FtsWI in an extended conformation poised for activation, with FtsN^E functioning as a tether between FtsI and FtsL to impact the interactions between the anchor-loop of FtsI and the catalytic cavity of FtsW, suggesting an activation mechanism of FtsWI by FtsN.

## Results

### Single-molecule tracking suggests that FtsQLB remains in complex with FtsWI on both FtsZ and sPG tracks

Previously, using single-molecule tracking, we showed that FtsWI exhibits two directionally moving subpopulations that reflect their activities in sPG synthesis:[17,30] a fast-moving subpopulation (~30 nm/s) driven by FtsZ treadmilling dynamics[30,31] and inactive in sPG synthesis (on the Z track) and a slow-moving subpopulation (~8 nm/s) independent of FtsZ treadmilling and driven by active sPG synthesis (on the sPG track). We later observed that processive FtsN exhibits only a slow-moving population, indicating association with FtsWI in an sPG synthesis complex on the sPG track[18]. To investigate whether FtsQLB is part of the processive, active complex on the sPG track and/or part of the inactive complex on the Z track, we ectopically expressed a Halo-FtsB fusion protein that complemented an FtsB depletion strain (Fig. S1A, Tables S1 and S2). We sparsely labeled Halo-FtsB with a fluorescent dye, JF646, and performed single-molecule tracking as we previously described[17].

We observed that a large percentage of single Halo-FtsB molecules exhibited directional motion (53.7 ± 2.7%, μ ± s.e.m., n = 379 segments) (Fig. 1B, top, Movie S1, and Movie S2). The velocity distribution of directional segments of Halo-FtsB single-molecule trajectories was best described as the sum of fast- and slow-moving subpopulations ($v_{fast} = 32.9 \pm 5.2$ nm/s, $v_{slow} = 8.2 \pm 1.7$ nm/s, $p_{fast} = 38\% \pm 13\%$, μ ± s.e.m., n = 179 segments, Fig. 1B, bottom, S1B, and Table S3). In the presence of fosfomycin, a drug that inhibits the synthesis of FtsW substrate Lipid II, Halo-FtsB shifted to the fast-moving subpopulation ($v_{fast} = 28.5 \pm 2.9$ nm/s, $v_{slow} = 8.5 \pm 3.0$ nm/s, $p_{fast} = 70\% \pm 10\%$, μ ± s.e.m., n = 218 segments, Figs. 1B, S1C, and

Table S3). The existence of two subpopulations of FtsB and the response to fosfomycin (Figs. 1B, S1B, S1C, and Table S3) mirrored observations for FtsW in our previous study[17], but differed from those of FtsN, which exhibited only slow motion[18]. Since FtsQ, FtsL, and FtsB form a stable heterotrimer[32], these observations suggest that FtsQLB is in complex with both fast-moving, inactive FtsWI on the Z track[17] and slow-moving, active FtsWI on the sPG track, with the latter also including FtsN[18]. As such, FtsWI in complex with FtsQLB may be able to adopt both active and inactive states, which are regulated by FtsN.

### An FtsQLBWI model describes an extended conformation of FtsI and extensive protein–protein interfaces

To gain insight into how the conformation of FtsWI can adopt both active and inactive states for sPG synthesis in the FtsQLBWI complex, we used the ColabFold[33] implementation of AlphaFold2 (AF2)[23] to predict the atomic structure of FtsQLBWI. In the prediction, we did not utilize template coordinates, avoiding explicit dependence on homologous published structures. The predicted structure of the complex showed an extended conformation of FtsI supported by interactions between the membrane-distal C-terminal domains of FtsQ, FtsL, FtsB, and FtsI, as well as interactions between transmembrane helices (TMH) of FtsL, FtsB, FtsW, and FtsI. These interactions were predicted with high local confidence (pLDDT) except for extreme terminal residues that have no predicted interactions (Fig. S2A). Furthermore, predicted DockQ values[34] for protein–protein interfaces with individual subunits ranged from 0.64 to 0.73 (Table S4), indicating accurate predictions of protein complexes[27]. Finally, predictions for homologous complexes of FtsQLBWI in other diderm (gram negative) and monoderm (gram positive) species were largely similar, with notable differences at less conserved C-terminal domains, such as the different interactions reported for PASTA domains in FtsI homologs *Bacillus subtilis* Pbp2B[35] and *Streptococcus pneumonia* Pbp2X[36] (Fig. S2B). These results suggest that AF2 can accurately predict protein–protein interfaces of *E. coli* FtsQLBWI and homologous complexes, but it remains essential to verify that predictions are consistent with experimental data and predictive in new experiments.

Next, we built a system of FtsQLBWI in a lipid bilayer (simulation details in Table S5) and performed a 1 μs all-atom molecular dynamics (MD) simulation to investigate the stability of predicted protein–protein interfaces and conformational dynamics of the complex. We excluded cytoplasmic, N-terminal regions of FtsQ (residues 1–19), FtsW (1–45), and FtsI (1–18) on the basis of having very low pLDDT and/or lacking high-confidence predicted protein–protein interactions. The MD simulation revealed a dynamic complex, with the FtsI transpeptidase (TPase) domain tilting backward and the head domain wrapping around the FtsL helix (Figs. 1C, S3 and Movie S3). Unless otherwise noted, we describe and depict structures after 1 μs of MD hereafter. Throughout the complex, we observed extensive interactions reported in previous experimental studies, such as leucine-zipper-like interactions between FtsL and FtsB helices[37] (Fig. S4A), and interactions between the same FtsW and FtsI transmembrane helices observed for the paralog RodA-PBP2[38,39] (Fig. S4B). The resulting complex is also consistent with crystal structures of *E. coli* FtsI (Fig. S5A)[40] and a partial FtsQ-FtsB complex (Fig. S5B)[41]. Finally, our model of FtsQLBWI and its dynamics during 1 μs of MD are largely consistent with a recently available Cryo-EM model of the orthologous complex in *P. aeruginosa*[42], which we discuss further in detail below.

To describe observed new interfaces in the context of previously defined domains, we define four periplasmic interaction regions in FtsQLBWI (Fig. 1C, dashed boxes). These include a Truss region that links FtsQ to FtsI via an extended β sheet, a Hub region that contains a dense interaction network on both sides of FtsL and FtsB helices, a Lid region in which the anchor domain of FtsI interacts with FtsW ECL4 (extracellular loop 4 containing the putative catalytic residue FtsW^D297,[43]), and a Pivot region where the first N-terminal periplasmic

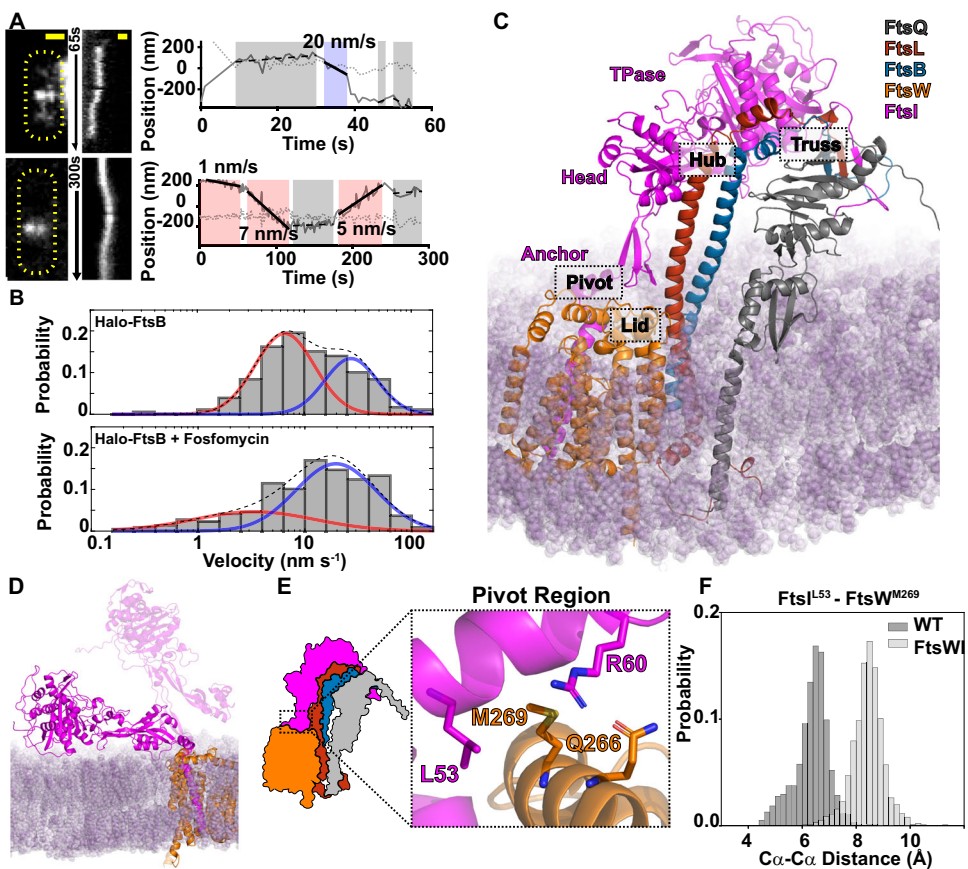

**Fig. 1 | Characterization and modeling of the *E. coli* FtsQLBWI complex.**
**A**, **B** Single-molecule tracking of Halo-FtsB suggests that FtsQLB remain in complex with FtsWI on both the fast-moving FtsZ-track and the slow-moving sPG synthesis track. **A** Two representative Halo-FtsB expressing cells with the maximum fluorescence intensity projection images (left), kymographs of fluorescence line scans at the midcell (middle), and unwrapped one-dimensional positions of the corresponding Halo-FtsB molecule along the circumference (solid gray line) and long axis (dotted gray line) of the cell were shown. Measured velocity of each segment and the corresponding classification (fast-moving, cyan; slow-moving, pink; stationary, gray) are labeled in the trajectory panels. Scale bar 500 nm. Similar images were observed in $N > 100$ cells. **B** Distribution of velocities of single Halo-FtsB molecules exhibiting directional motion in wild-type *E. coli* cells grown in minimal media in the absence (top) of fosfomycin was best fit with two moving populations, one slow (red) and one fast (blue). A dashed line indicates the summed probability. In the presence of fosfomycin that inhibits cell wall synthesis (bottom), the slow-moving population (red) is drastically reduced. These dynamics behaviors are

similar to those of FtsW or FtsI[17]. **C** Modeled structure of *E. coli* FtsQLBWI within a POPE bilayer (purple) in the last frame of a 1-µs MD simulation. The complex consists of FtsQ 20–276 (gray), FtsL 1–121 (red), FtsB 1–113 (blue), FtsW 46–414 (orange), and FtsI 19–588 (magenta). The FtsI TPase, head and anchor domains are labeled in magenta text. The four interface regions—Pivot, Truss, Hub, and Lid—are highlighted in dashed boxes. **D** In the absence of FtsQLB, FtsI (magenta) collapses to the membrane (purple) at the end of the 1-µs MD simulation. The position of FtsI at the beginning of the simulation (transparent magenta) is shown for comparison. **E** Zoomed-in view of the Pivot region in FtsQLBWI, in which interactions between FtsI[L53] (magenta) and FtsW[M269] (orange) and between FtsI[R60] (magenta) and FtsW[Q266] (orange) secure the position of the FtsI anchor domain (magenta) with respect to FtsW (orange). **F** In the absence of FtsQLB, interactions between FtsI[L53] and FtsW[M269] are broken, as shown by the increased Cα-Cα distances between the two residues (light gray) compared to that in the presence of FtsQLB (WT, dark gray) in the last 500 ns of the MD simulation. Source data are provided as a Source Data file.

helix of FtsI interacts with FtsW ECL4. FtsI is involved in all four regions, contacting FtsL and FtsB in the membrane-distal periplasmic space and FtsW in the membrane, but essentially has no interaction with FtsQ. These four regions are rich in residues that, when mutated, give rise to superfission (SF) or dominant negative (DN) phenotypes, suggesting that they modulate FtsWI activity. While our analysis below focuses on a subset of residues and protein–protein interfaces with well-established phenotypes, we provide trajectories of top-ranked inter-residue hydrogen-bonding frequencies for each interface pair in the complex (Supplementary Data 1).

**A collapsed structural model of FtsWI in the absence of FtsQLB reveals critical interactions in the Pivot region**
Our FtsQLBWI model and FtsW, FtsN, and FtsB single-molecule tracking results suggest that FtsQLB forms a complex with FtsWI on the Z track that is poised for further activation by FtsN on the sPG track. To investigate this possibility, we performed an MD simulation of an FtsWI

system without FtsQLB. In this simulation we observed flexibility in FtsI periplasmic domains in the absence of FtsQLB. FtsI rotated about the Pivot region and collapsed onto the membrane, where it remained until the end of the 1 µs simulation (Fig. 1D and Movie S4). While 1 µs is insufficient to sample all conformations of FtsI in equilibrium, a collapsed FtsI conformation is consistent with various flexible conformations of the paralogous *Thermus thermophilus* PBP2 in the elongation complex with RodA in a previous cryo-EM study[39]. Notably, FtsI tilts in the opposite direction relative to FtsW in comparison to a RodA-PBP2 crystal structure, with *E. coli* FtsI missing a small loop that mediates RodA-PBP2 interaction in *T. thermophilus* PBP2 (Fig. S5C).

In the FtsQLBWI structure, the Pivot region consists of an FtsI helix (FtsI[D51–S61]) on top of the second short helix (FtsW[S260–G274]) of FtsW ECL4 and is maintained by hydrogen bonding between FtsI[R60]-FtsW[Q266] (Fig. S6A) and hydrophobic contact between FtsI[L53] and FtsW[M269] (Figs. 1E and S7A). These contacts were broken in FtsWI in the absence of FtsQLB, resulting in separation of FtsI[L53] and FtsW[M269] and of FtsI[R60]

and FtsW$^{Q266}$ in the FtsWI trajectory (Figs. 1F, S6B, and S7B). Previous work showed that mutations to charged residues in this interface (FtsI$^{G57D}$, FtsW$^{M269K}$) lead to the failure of cell wall constriction, while replacing with a hydrophobic residue (FtsW$^{M269I}$) produced a SF variant of FtsW[20,21,44]. As the Pivot region conformation is impacted by distal interactions with FtsQLB and contains both DN and SF mutations, it is likely key for regulating FtsWI activity.

### A β-sheet formed by the extreme C-termini of FtsQLBI in the Truss region stabilizes an extended conformation of FtsI

To identify what specific interactions between FtsI and FtsQLB support the extended conformation of FtsI periplasmic domains, we examined the Truss region. The Truss region consists of the C-terminal domains of FtsQLB and FtsI, with a striking feature that each one of the four proteins contributes one β-strand to extend an FtsQ β-sheet (Figs. 1C

and 2A). The β-sheet formed between FtsQ$^{A252-A257}$ (β12) and FtsB$^{T83-P89}$ (β1) has been previously reported[45] (Fig. S5B), but that between FtsL$^{N116-Q120}$ (β1) and FtsI$^{E575-I578}$ (β16) has not been observed experimentally and was only recently identified by structure prediction[46]. FtsL indirectly interacts with FtsQ via FtsB within this β-sheet. The β-strand of FtsL$^{N116-Q120}$ maintains β dihedral angles throughout the 1-μs simulation, while FtsI$^{E575-I578}$, which terminates the β-sheet, exhibits greater flexibility (Fig. S8A, B). Previously, it was reported that the FtsI C-terminus (FtsI$^{I578-S588}$) was cleaved, although it was unclear whether the cleavage occurred post-translationally, or during sample processing[47,48]. We performed an additional 200-ns MD simulation of FtsQLBWI$^{ΔI578-S588}$ and observed that this truncation did not disrupt β-strand formation of FtsL$^{N116-Q120}$ or FtsI$^{E575-V577}$ (Fig. S8C, D). In addition to the β-sheet linking these four proteins, two important contacts were observed between FtsI$^{R559}$ and FtsL$^{E115}$, and between FtsI$^{R239}$ and

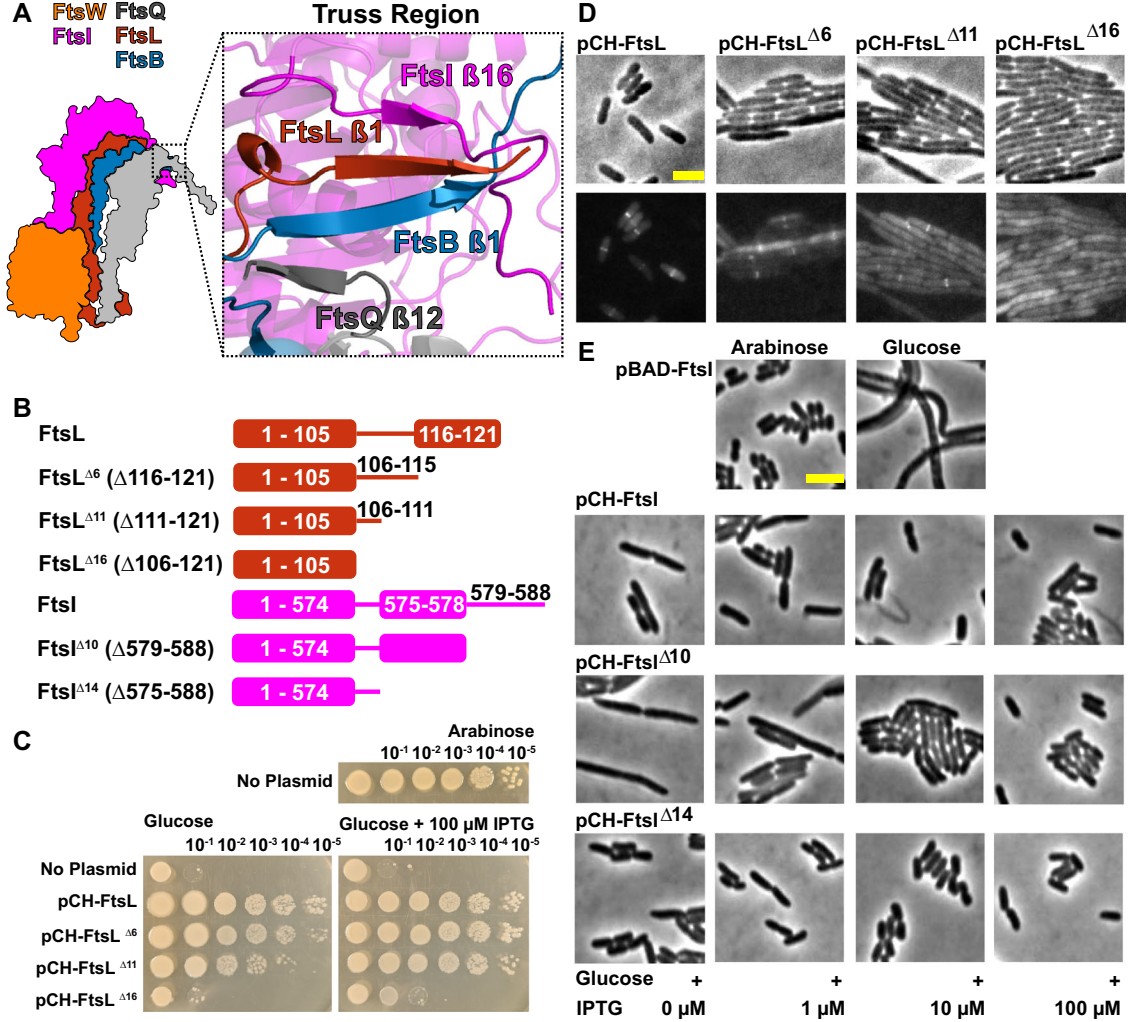

**Fig. 2 | C-terminal extended β-sheet of FtsQLBI in the Truss region is important for cell division. A** A detailed view of the Truss region in the final frame of the FtsQLBWI 1-μs MD simulation illustrates β-sheet interactions between the C-terminal ends of FtsQ (gray), FtsB (blue), FtsL (red), and FtsI (magenta). **B** Cartoon showing FtsL and FtsI β-strand truncation mutants. Also see Fig. S12 for details. **C** Spot dilution complementation test of FtsL truncation mutants. *E. coli* cells depleted of chromosomal wild-type FtsL but contain FtsL expressed from the P$_{BAD}$ promoter (strain MDG279, Table S2) complemented in the presence of arabinose (top panel), but failed in the presence of glucose (No Plasmid, first rows of the bottom panel). The same depletion strain (MDG279) expressing wild-type FtsL (pCH-FtsL, or pBMB064), FtsL$^{Δ6}$ (pCH-FtsL$^{Δ6}$, or pBMB065) and FtsL$^{Δ11}$ (pCH-FtsL$^{Δ11}$, or pBMB066, Table S1) from a *lac* promoter on plasmids complemented the

depletion in the presence of glucose at both no induction and 100 μM IPTG conditions (middle rows of both panels). pCH-FtsL$^{Δ16}$ is unable to complement at both conditions (bottom rows of the bottom panel). See Fig. S13 for more induction conditions. **D** Images of *E. coli* cells depleted of wild-type FtsL and expressing an mVenus fusion to FtsL of various truncations showed that truncations of FtsL of increasing length exhibited increasing cell length (top) and decreased FtsL midcell localization (bottom) relative to cells expressing mVenus fused to full-length FtsL. Scale bar 3μm. See Fig. S13 for quantifications. **E** Images of *E. coli* cells depleted of wild-type FtsI and expressing FtsI truncations. A wild-type FtsI fusion and FtsI$^{Δ14}$ exhibit near-normal cell lengths even at low induction levels, while FtsI$^{Δ11}$ exhibits filamentous cells at low expression levels. Scale bar 3 μm. See Fig. S14 for quantifications. Source data are provided as a Source Data file.

FtsQ[D249], which may strengthen C-terminal interactions between FtsI and FtsQLB (Figs. S9, S10, and S11).

To verify the importance of the β-sheet formed between FtsL and FtsI, we constructed three FtsL mutants with varied degrees of C-terminal truncations (FtsL[Δ6], FtsL[Δ11], and FtsL[Δ16]). All three lack the C-terminal β-strand, and FtsL[Δ11] and FtsL[Δ16] additionally lack residues such as FtsL[E115] that interacts with FtsI (Figs. 2B and S12). FtsL[Δ6] and FtsL[Δ11] cells were filamentous, indicating a division defect, but complemented an FtsL depletion strain at the highest induction level tested (100 μM IPTG). FtsL[Δ16] cells remained filamentous and failed to rescue cell division under the same condition (Figs. 2C, S13A, B, and Table S6). Fluorescently labeled FtsL[Δ6] and FtsL[Δ11] (mVenus-FtsL) fusions showed clear, but significantly reduced, midcell localization, while mVenus-FtsL[Δ16] cells exhibited only diffusive, cytoplasmic fluorescence (Figs. 2D, S13C, Tables S7, and S8). These different localization patterns were not due to mVenus-FtsL truncations' differential expression levels or their abilities to support cell division (Fig. S13D, E). These results suggest that the interactions of the C-terminal β-strand of FtsL with FtsB and FtsI are important for the recruitment of the complex, while further truncations impact essential divisome functions. They are consistent with a previous observation that a C-terminal truncation mutant, FtsL[Δ114–121] (or FtsI[Δ7]), complemented FtsL depletion, but was defective in co-immunoprecipitating with FtsQ[49] and produced wrinkled colonies[50].

Next, to investigate the role of β-strand interaction between FtsL and FtsI, we constructed two FtsI C-terminal truncations, FtsI[ΔN579–S588] (or FtsI[Δ10]), where only the disordered C-terminus after the predicted β-strand was deleted, and FtsI[ΔE575–S588] (or FtsI[Δ14]), where the β-strand and the C-terminus were both deleted (Figs. 2B, S12, and Table S9). We note that reported post-translational cleavage of FtsI between FtsI[I578–S588] would be FtsI[Δ11] following this nomenclature, removing one residue with a hydrophobic sidechain from the predicted β-strand that interacts with FtsL. Both FtsI[Δ10] and FtsI[Δ14] complemented FtsI depletion (Fig. S14A). FtsI[Δ10] cells were longer ($l = 5.6 \pm 1.3$, $\mu \pm$ s.e.m. calculated from two biological replicates for $n = 446$ total cells) than wild-type cells ($l = 3.9 \pm 0.0$, $\mu \pm$ s.e.m., $n = 1215$ cells) at the minimum expression level (1 μM IPTG). However, extending the truncation in FtsI[Δ14] corrected this defect, resulting in cell lengths marginally shorter than wild-type ($l = 3.3 \pm 0.0$, $\mu \pm$ s.e.m., $n = 487$ cells) at the minimum expression level (Figs. 2E, S14B, and Table S9). Western blot showed that FtsI[Δ10] expression was lower than that for either FtsI or FtsI[Δ14] at equivalent induction levels, and that FtsI[Δ14] expression was marginally higher than that of FtsI (Fig. S14C, D and Table S10). Little change was observed in midcell localization of FtsI across all conditions (Fig. S14E and Table S11). Together, these observations indicate that interactions between C-terminal β-strands of FtsI and FtsL may modulate FtsI stability and/or FtsWI activity.

## Hub region reveal inhibitory and activating interactions between FtsL and FtsI

The Hub region lies beneath the Truss region, encompassing the previously identified CCD interface (Constriction Control Domain, containing multiple SF residues[9,10,51]) on one side of FtsL and FtsB helices (Figs. 3A and S15), and the AWI interface (Activation of FtsW and FtsI, containing multiple DN residues[52]) on the other side (Fig. 3B). As such, the Hub region may be a key regulatory region for the activities of FtsWI. We analyzed predicted interactions in the Hub region that were maintained during the final 500 ns of a 1 μs simulation of the FtsQLBWI system, focusing on hydrogen bonds (as defined in "Methods") and hydrophobic contacts by computing the distance between the side chain geometry center of each residue in a given pair.

The most prominent feature in the CCD interface of the Hub region is a network of residues with polar sidechains extending through FtsQLB to FtsI. As shown in Fig. 3A, contacts between FtsQ[R196]-FtsB[E69], FtsQ[R196]-FtsB[E56] and FtsQ[R213]-FtsB[E56] connect FtsQ to FtsB. FtsB is

connected to FtsL through FtsB[E56]-FtsB[R70] and FtsB[R70]-FtsL[E88]. Finally, FtsL connects to FtsI through FtsL[R82]-FtsI[S85] (Figs. 3A and S16). Mutations of residues in this region such as FtsB[E56A], FtsL[E88K], FtsL[N89S], FtsL[G92D], and FtsL[H94Y] have been reported to be SF variants; cells expressing these variants are shorter than wild-type cells and able to survive in the absence of FtsN[10,15]. Therefore, it is possible that these interactions maintain FtsI in an inactive conformation, and that abolishing these inhibitory interactions activates FtsI.

Interactions in the AWI interface are mainly between one face of the FtsL helix and two β-strands at the neck of the FtsI head domain (Fig. 3B). A few hydrophobic residues, FtsI[Y168], FtsI[V84], and FtsI[V86], pack closely with hydrophobic residues FtsL[I85] and FtsL[L86]. Additionally, hydrogen bonds were observed between FtsL[R82]-FtsI[S85] and between FtsL[N83]-FtsI[P87] (Fig. S16, hydrogen-bond frequencies for all protein–protein interfaces in the complex are available in Supplementary Data 1). Previously both FtsL[R82E] and FtsL[L86F] were identified as DN mutants[10], suggesting that these interactions may be required for maintaining the active conformation of FtsI.

## MD simulation of the FtsI[R167S] SF variant suggests an active conformation of FtsI

We reasoned that if predicted inhibitory interactions in the CCD region and activating interactions in the AWI region play important roles in modulating FtsI activity, we may observe corresponding conformational changes in the Hub region when some of these residues are mutated. To examine interactions in the AWI region, we performed a 1-μs MD simulation of a SF variant, FtsI[R167S][17] (Fig. S17A). FtsI[R167] is next to hydrophobic residues FtsI[Y168], FtsI[V84], and FtsI[V86] that interact with FtsL[I85] and FtsL[L86] (Fig. 3B). In the FtsI[R167S] simulation, the two β-strands at the neck of the FtsI head domain pack closer to FtsL than in the WT (FtsQLBWI) simulation, as measured by the sidechain geometry center distances between FtsI[V84] and FtsL[I85] (Figs. 3C, D, and S18A, B; $d = 6.4 \pm 0.6$ Å for WT and $5.4 \pm 0.5$ Å for FtsI[R167S] complex). Consistent with this measurement, the solvent-accessible surface area of FtsL residues with hydrophobic sidechains in the helix linking the Lid and Hub regions was reduced from 534 Å² to 501 Å².

Interestingly, we observed drastic conformational changes in the turn connecting the two FtsL helices immediately above the AWI region of the Hub. As shown in Fig. 3E, this conformational change corresponds to a transition in the distribution of dihedral angles for FtsL[G92] and with FtsL[H94] shifting from α-helical to β-strand dihedral angles, while the WT complex retains α-helical dihedral angles. Note that both FtsL[H94Y] and FtsL[G92D] are SF variants, suggesting that the conformation of this region plays an important role in FtsWI regulation. Taken together, the FtsI[R167S] SF complex simulation suggests that replacing the charged arginine residue with uncharged serine in FtsI[R167S] strengthens hydrophobic interactions between FtsI and FtsL in the AWI region and results a conformational change in FtsL helices. These changes may correspond to an activated conformation of FtsWI, leading to the SF phenotype of FtsI[R167S].

## FtsB remains associated with FtsWI in the FtsI[R167S] SF background

In our previous single-molecule tracking experiments, we observed that FtsWI shifts to the slow-moving, active population on the sPG track in the FtsB[E56A], FtsI[R167S], and FtsW[E289G] SF backgrounds[17]. We asked whether activation of FtsWI was due to the dissociation of FtsQLB from FtsWI on the Z track to relieve an inhibitory effect as previously proposed[19], or if FtsQLB remains in complex with active FtsWI on the sPG track to maintain its activities as recently suggested by an in vitro study[9]. To distinguish between these two possibilities, we performed single-molecule tracking of Halo-FtsB in the FtsI[R167S] SF background under a rich growth condition, which was previously used to assess the effect on FtsW. As shown in Fig. 3F, we observed

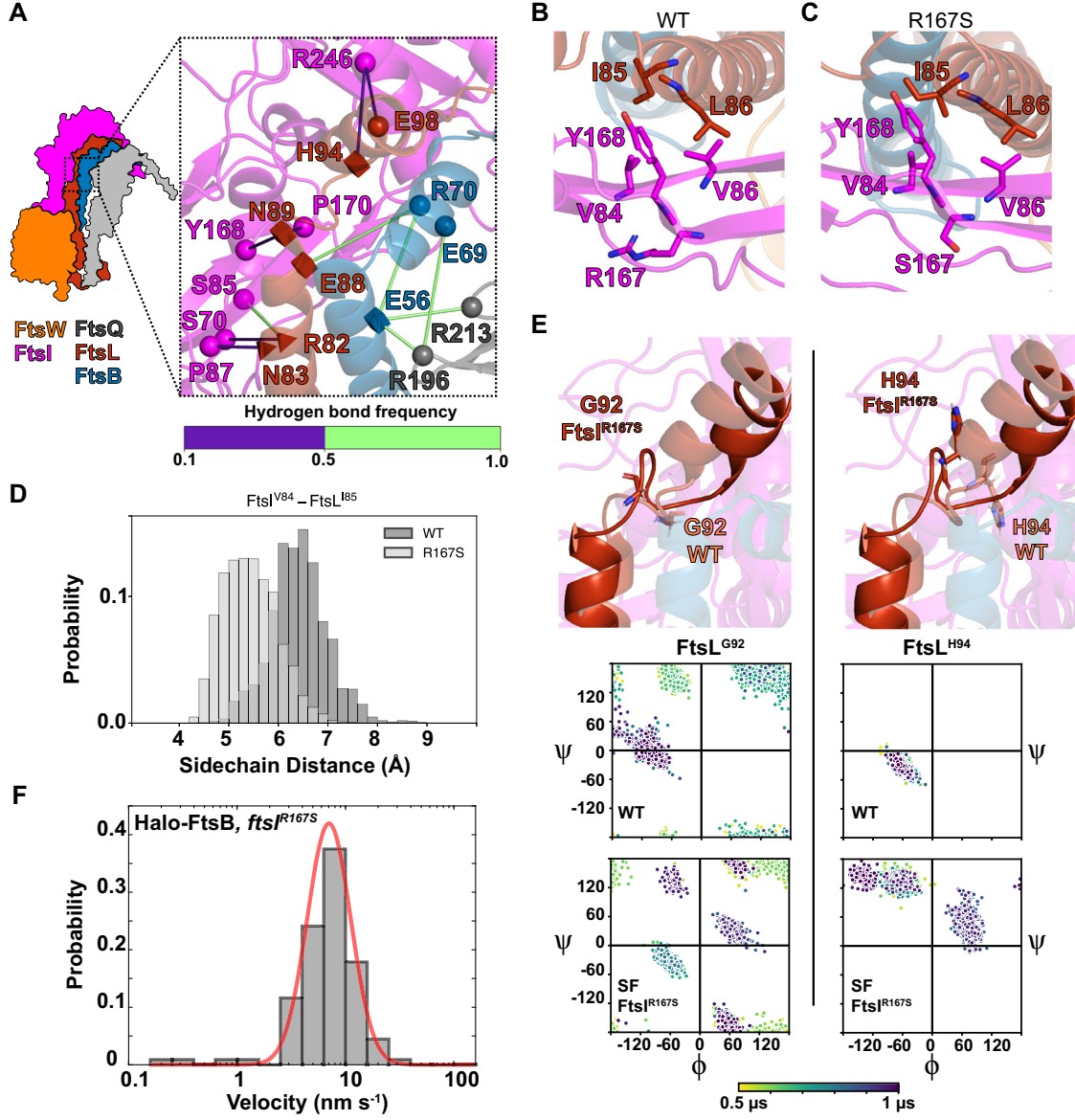

**Fig. 3 | Inhibitory and activating interactions between FtsL and FtsI in the Hub region are distributed along the CCD (A) and AWI interfaces (B and C). A** A hydrogen-bonding network extends from FtsQLB to FtsI in the CCD interface of the Hub region. Shapes indicate residues involved in hydrogen bonding. Superfission mutant residues (FtsL$^{N89}$, FtsL$^{E88}$, FtsL$^{H94}$, and FtsB$^{E56}$) are depicted as squares. Dominant negative mutant residues (FtsL$^{R82}$ and FtsL$^{N83}$) depicted as tetrahedrons. Spheres indicate other residues involved (FtsI$^{P87,S70,S85,Y168,P170,R246}$, FtsL$^{E98}$, FtsB$^{E69,R70}$, and FtsQ$^{R196,R213}$). Hydrogen-bond frequencies during the last 500 ns of FtsQLBWI simulation are indicated by the color bar. **B** Hydrophobic packing among FtsL$^{I85}$, FtsL$^{L86}$, FtsI$^{V86}$, FtsI$^{V84}$, and FtsI$^{Y168}$ is observed in the AWI domain between FtsL (red) and FtsI (magenta) in the WT FtsQLBWI simulation. **C** The same AWI interface exhibits a tighter hydrophobic packing in the FtsQLBWI$^{R167S}$ complex simulation. Note the relative differences between FtsI$^{V84,Y168}$ and FtsL$^{I85}$. **D** The distance between the centers of geometry of FtsI$^{V84}$ and FtsL$^{I85}$ sidechains decreased from $d = 6.4 \pm 0.6$ Å in the WT FtsQLBWI complex to $5.4 \pm 0.5$ Å in the FtsQLBWI$^{R167S}$ in the

last 500 ns of simulations, consistent with the closer packing between the two residues in the SF FtsQLBWI$^{R167S}$ complex. **E** Top: conformations after 1 µs MD of FtsL$^{G92}$ (left) and FtsL$^{H94}$ (right) in the WT FtsQLBWI (tan color) and the SF FtsQLBWI$^{R167S}$ (dark red color) complexes show disruption of the second short α-helix of FtsL in the FtsQLBWI$^{R167S}$ complex. Bottom: dihedral angles of both FtsL$^{G92}$ (left) and FtsL$^{H94}$ (right) are disrupted in the FtsQLBWI$^{R167S}$ complex. A shift for FtsL$^{H94}$ dihedral angles was observed from being α-helical (bottom left quadrant) for WT FtsQLBWI to being β-strand-like in FtsQLBWI$^{R167S}$ (top left quadrant). Dihedral angles are color-coded from yellow to dark purple in time in the 1 µs MD trajectories. **F** Single-molecule tracking of Halo-FtsB in a strain expressing FtsI$^{R167S}$ shows that Halo-FtsB molecules only exhibit the slow-moving population. The velocity histogram (gray bars) is best fit by a single population (red, $v_{slow} = 8.0 \pm 0.4$ nm/s, µ ± s.e.m., $n = 112$ segments). Source data are provided as a Source Data file.

that Halo-FtsB was best fit as a single, slow-moving, active population ($v_{slow} = 8.0 \pm 0.4$ nm/s, µ ± s.e.m., $n = 112$ segments, Table S3), just as we previously observed for FtsW. This result is consistent with the expectation that FtsI$^{R167S}$ enhances the formation of the active FtsQLBWI complex on the sPG synthesis track, and that FtsQLB remains in the complex with activated FtsWI, as we previously reported for FtsN[18].

## Lid region reveals important interactions between FtsI anchor domain and FtsW ECL4 that may modulate FtsW activity

The Lid region is located at the periplasmic face of the inner membrane and involves interactions between a polar patch on the back of FtsL and FtsB[24], a loop in the FtsI anchor domain (FtsI anchor-loop) and FtsW ECL4 (Figs. 4A, S19, and S20). Specifically, hydrogen bonds between FtsB$^{K23}$-FtsI$^{E219}$, FtsB$^{D35}$-FtsL$^{R67}$, FtsL$^{R67}$-FtsI$^{D220}$, and FtsL$^{E68}$-

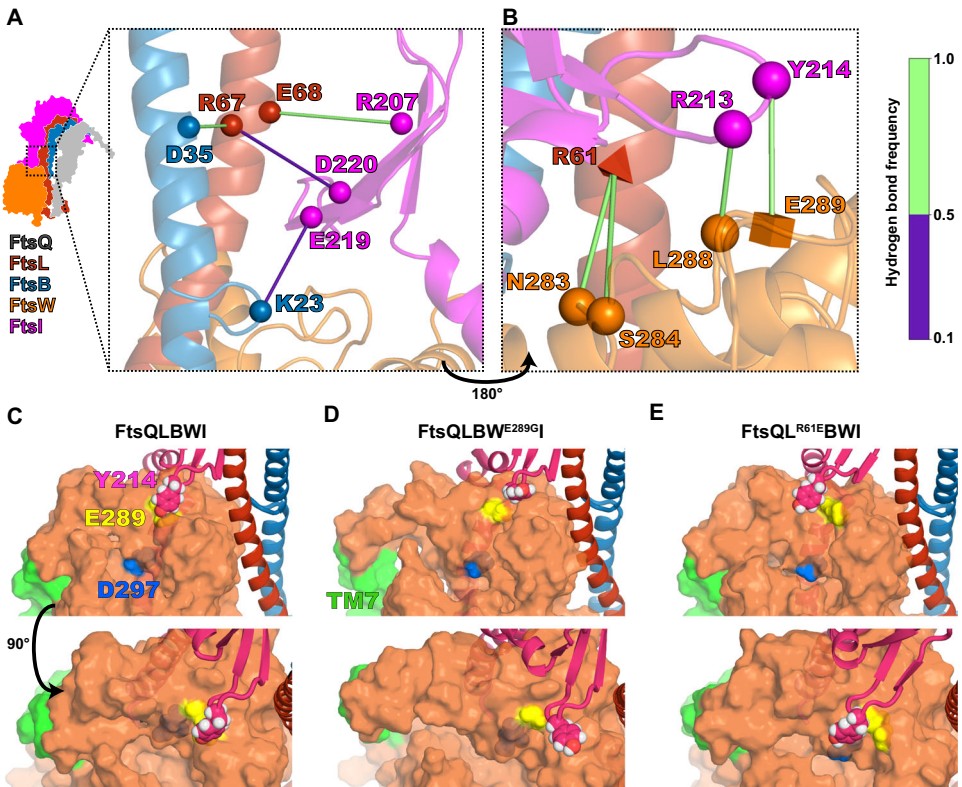

**Fig. 4 | Interactions between FtsB, FtsL, FtsI, and FtsW position the FtsI anchor-loop near the FtsW active site; FtsI anchor-loop position is correlated with FtsWI activity. A, B** Hydrogen bonding in the Lid region. Shapes indicate residues with hydrogen bonding during the last 500 ns of FtsQLBWI simulation, with a cube specifying SF mutation FtsW[E289G], a tetrahedron specifying DN mutation FtsL[R61E], and spheres indicate other residues involved (FtsI[R207, R213, Y214, E219, D220], FtsL[R67, E68], FtsB[K23, D35]). Hydrogen-bonding frequencies during the last 500 ns of the simulation are indicated by the color bar. Two views of the Lid region are shown: **A** interactions between the FtsI anchor domain (magenta), FtsL (red), and FtsB (blue), which position the anchor domain above FtsW ECL4; **B** the same region rotated ~180° to show how interactions between residues in ECL4 of FtsW (orange) with FtsL[R61],

FtsI[R213], and FtsI[Y214] position FtsW ECL4 (orange) below the FtsI anchor-loop (magenta). **C–E** Side and top views of a cavity on the periplasmic face of FtsW (orange) containing the putative catalytic residue FtsW[D297] (blue) lying below the FtsW[E289] residue (yellow; **D** shows SF FtsW[E289G]) for WT FtsQLBWI (**C**), SF variant FtsQLBW[E289G]I (**D**) and DN mutant FtsQL[R61E]BWI (**E**) complexes following 1 μs MD simulations. With SF mutation FtsW[E289G], the FtsI anchor-loop (magenta) including FtsI[Y214] (pink and white spheres) moves away from the cavity, which is expanded as FtsW TM7 (green) tilts into the bilayer. With DN mutation FtsL[R61E], the anchor-loop (magenta) moves and FtsI[Y214] moves over the cavity, which is stabilized by an interaction between FtsI[R216] and FtsW[E289]. Source data are provided as a Source Data file.

FtsI[R207] persist during the final 500 ns of FtsQLBWI simulation and fix the orientation of the FtsI anchor-loop with respect to the FtsL and FtsB helices (Figs. 4A and S19), while hydrogen bonding between FtsL[R61], FtsW[N283], and FtsW[S284] positions ECL4 directly beneath the anchor-loop (Figs. 4B and S20). These interactions allow FtsI[Y214] in the FtsI anchor-loop to hydrogen bond with FtsW[E289], positioning FtsW ECL4 beside a central pore containing the putative catalytic residue FtsW[D297] [20] (Figs. 4B, C, and S20). Previous studies showed that FtsW[E289G] is a SF variant while FtsL[R61E] and FtsL[R67E] are DN variants [24], indicating that altering the interactions in the Lid region can indeed render a constitutively active or inactive complex.

Observing that Lid region interactions in FtsQLBWI are rich in SF and DN residues, we hypothesize that local conformational changes resulting from these mutations could shed light on the FtsW activation mechanism. We simulated the effects of introducing the FtsW[E289G] SF variant and the FtsL[R61E] DN variant into FtsQLBWI. Both complexes adopt similar global structures after 1 μs of MD compared to WT FtsQLBWI (Fig. S17B, C), but differ in local positioning of the FtsI anchor-loop relative to FtsW ECL4 (Fig. 4D, E). In the SF FtsW[E289G] complex, the FtsI anchor-loop is rotated sideways toward FtsL and FtsW TMH1, opening a central cavity of FtsW (Fig. 4D). This conformation is stabilized by backbone hydrogen bonding between FtsI[R213] and FtsW[L288] in addition to maintaining contacts with FtsL (Fig. S20). In contrast, in the DN FtsL[R61E] complex, the FtsI anchor-loop is directly

above the FtsW cavity (Fig. 4E). This interaction is stabilized by contacts between FtsI[R216] and FtsW[E289] and persists through ~72% of the last 500 ns of simulation of the FtsL[R61E] DN complex (Fig. S21). As we described above, the FtsI anchor-loop is coordinated by its interactions with the polar patch on one side of FtsL helix and with FtsW ECL4, which are reduced in the FtsL[R61E] variant (Fig. S17C). Loss of the interaction between FtsL[R61E] and FtsI[E219] is also accompanied by conformational changes in FtsI; the relative orientation of FtsI and FtsW helices in the Pivot region changes, and the β-sheet in the anchor domain tilts towards FtsL (Fig. S17C). We also observed that in the SF FtsW[E289G] simulation FtsW TM7 tilts outward and expose a large extracytoplasmic cavity (Figs. 4D and S17B) as reported for the RodA-PBP2 complex [39]. In simulations of both FtsWI in the absence of FtsQLB and FtsQLBWI with FtsL[R61E], FtsW[Y379] in ECL5 flipped from its predicted position to fit between FtsW TM1 and TM2. This flip was only observed in simulations of these inactive complexes, suggesting that FtsW-FtsL interactions may influence conformation near the FtsW active site.

### Binding of FtsN[E] reduces inhibitory interactions and strengthens activating interactions in the FtsQLBWI complex

Previous studies have shown that the effects of SF variants are independent of FtsN, *i.e.* SF variants bypass the need for FtsN binding. Therefore, we reason that binding of FtsN to FtsQLBWI could switch the complex into an active conformation similar to what we observed

for SF variants. To investigate the role of FtsN binding, we predicted the structure of FtsQLBWI with the addition of FtsN[K58–V108], which encompasses FtsN[E] (FtsN[L75–Q93])[15] and a sufficient number of surrounding periplasmic residues required to obtain a predicted structure. As with other complexes, we performed 1 μs of MD to investigate interactions and dynamics of the predicted complex of FtsQLBWI and FtsN[K58–V108] (FtsQLBWIN, Movie S5).

While termini of FtsN[K58–V108] did not form stable interactions on this timescale, the FtsN[E] region is bookended by two interfaces predicted with high confidence and stability in MD (Fig. 5A). First,

FtsN[L75–P79] forms a polyproline II helix that binds the FtsI head domain (Fig. S22A). This region consists of residues with high polyproline II helix propensity[53], which is a property conserved for FtsN (Fig. S22B) in taxonomic families within the previously identified subgroups characterized by a four-amino-acid deletion in RpoB[54,55]. However, we did not identify this motif for FtsN outside of these subgroups (*e.g.* in *Pseudomonadaceae*) and predicted complexes with both *P. aeruginosa* FtsI and PBP3x did not include this interaction (Fig. S22C), suggesting that FtsN interactions and functions may vary between γ-proteobacteria. This observation is consistent with the conditional

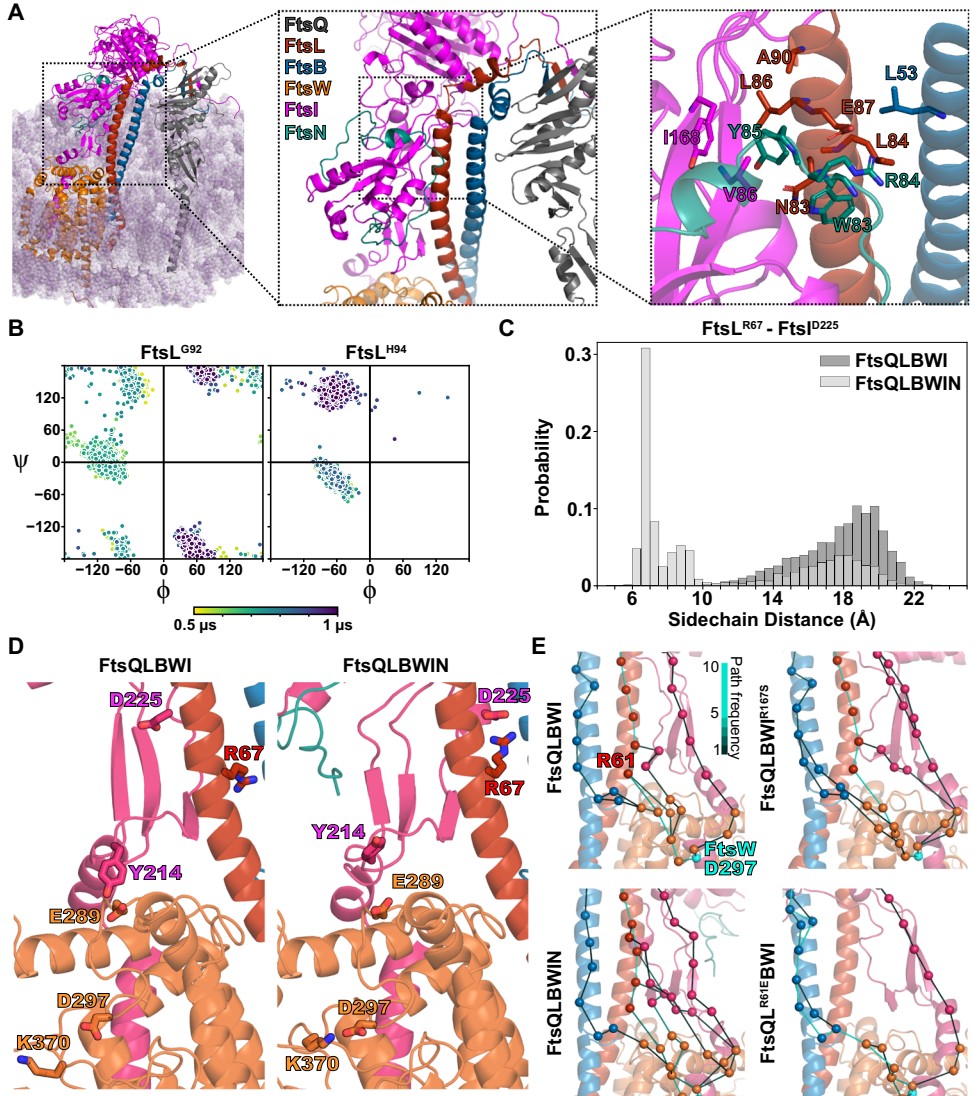

**Fig. 5 | FtsN[E] binding reduces inhibitory interactions and induces conformational changes observed in SF complexes. A** Conformer of the FtsQLBWIN complex after 1 μs MD, with FtsN[E] shown in teal and detailed views of FtsN binding in the Hub region. Highly conserved and essential residues of FtsN, FtsN[W83], FtsN[R84], and FtsN[Y85] are at the AWI interface in the Hub region, close to residues previously identified in the FtsI[R167S] SF variant complex, including FtsI[V86, I168] and FtsL[L86, N83, A90, E87, L84]. **B** Dihedral angle analyses of FtsL[G92] (left) and FtsL[H94] (right) in the FtsN-bound complex showed similar backbone dihedral angle changes compared to those in the SF mutant FtsQLBWI[R167S] (Fig. 3E). **C** The distribution of distances between sidechain centers of geometry for FtsL[R67] and FtsI[D225] in the FtsQLBWIN simulation (light gray) exhibited a second, decreased peak compared to that in the FtsQLBWI simulation (dark gray). The distance decrease in FtsQLBWIN occurred with a conformational change in the last 300 ns of the MD simulation (trajectory in Fig. S23), demonstrating tighter packing between the FtsI anchor domain and FtsL in the presence of FtsN. **D** Interaction between the FtsI anchor-

loop residue (magenta) FtsI[Y214] and FtsW[E289] (orange) is disrupted in FtsQLBWIN as FtsI[D225] in the anchor domain increased its interaction with FtsL[R67]. This interaction moves the sidechain of FtsI[Y214] (magenta) away from the putative catalytic residue FtsW[D297] (orange), similar to what was observed in the SF complex FtsQLBW[E289G]I (Fig. 4D). Note that FtsW[K370] forms a salt bridge with FtsW[D297] in FtsQLBWIN, but not FtsQLBWI. **E** Ensembles of optimal paths calculated from FtsQ to FtsW[D297] (teal) showing the 10 most optimal paths in the final 500 ns of MD for FtsQLBWI (top left), FtsQLBWI[R167S] (top right), FtsQLBWIN (bottom left), and FtsQL[R61E]BWI (bottom right). Line colors correspond to the number of paths connecting pairs of residues in the ensemble (colorbar). The DN variant FtsL[R61E] (bottom right) eliminated all direct paths between FtsL to FtsW, shifting to paths through a small loop extending from the FtsB helix. Addition of FtsN[E] increased the density of paths through FtsI, which also includes paths through the Pivot region. Source data are provided as a Source Data file.

essentiality of FtsN and the inability to detect an effect of FtsN on FtsQLBWI or reconstitute FtsQLBWIN in vitro for *P. aeruginosa*[9,42].

The second interaction region, FtsN[W83–L89], contains conserved residues FtsN[W83], FtsN[Y85], and FtsN[L89], which are also the most sensitive residues to alanine scanning mutagenesis[15]. This region interacts with both the AWI interface of FtsL (FtsL[L86], FtsL[A90]) and at the neck of the FtsI head domain (FtsI[V86] and FtsI[Y168]), largely through hydrophobic interactions and also including FtsB[L53] (Fig. 5A, right). Importantly, the FtsL and FtsI residues bound by FtsN overlap with those impacted in the FtsI[R167S] SF variant (FtsL[L86], FtsI[V86], and FtsI[Y168]), suggesting a shared mechanism of FtsQLBWI activation. Furthermore, this FtsN-binding interface includes both FtsL[A90] in the hydrophobic interface and an FtsL[E87]-FtsN[R84] interaction, suggesting that DN variants FtsL[E87K] and FtsL[A90E] are defective in FtsN[E] binding. The solvent-accessible surface area of FtsL residues with hydrophobic sidechains in this region was reduced from 534 Å² in WT to 383 Å², greater than that in the FtsI[R167S] simulation.

To validate the prediction of FtsN binding specificity, we additionally predicted structures of FtsN[K58–V108] together with only FtsQLB or FtsWI. Despite relatively low prediction confidence, the predicted location of FtsN[W83–L89] binding did not change (Fig. S22D). We also note that predictions of full-length FtsN are possible for *E. coli* and homologous complexes such as for *Aliivibrio fischeri*, but we failed to identify other high-confidence binding interfaces (Fig. S22E).

Little global change was observed in simulations of FtsQLBWIN compared to FtsQLBWI (Fig. S17D). However, MD revealed that the presence of FtsN[E] triggered extensive local conformational changes in all four interaction regions in FtsQLBWI. In the Truss region, the FtsL[E115]-FtsI[R559] interaction was broken (Figs. S9A and S10E), which impacts interactions between the FtsL β-strand and adjacent FtsB and FtsI β-strands. Several interactions involving polar residues at the CCD interface (FtsL[E98]-FtsI[R246] and FtsL[N89]-FtsI[P170]) in the Hub region were nearly completely abolished (Fig. S16), resulting in conformational change for FtsL[H94] and FtsL[G92] similar to that observed for FtsQLBWI[R167S] (Fig. 5B). In the Lid region, charged interactions between the polar patch of FtsL and the back of the FtsI anchor domain (FtsL[R67]-FtsI[D225], Figs. 5C and S23) are present; FtsW[E289] in ECL4 switches from interacting with FtsI[Y214] to FtsI[R213] in the anchor-loop, resulting in conformation changes in the FtsI anchor-loop and FtsW ECL4 that alter the catalytic cavity on the periplasmic face of FtsW relative to the conformer observed for FtsQLBWI (Fig. 5D). This conformational change, coincident with loss of FtsW[E289]-FtsI[Y214] interaction, is similar to that observed with the FtsW[E289G] SF mutation (Figs. 4D and S22F). However, the same conformational change did not occur in the FtsI[R167S] simulation within 1 μs and it remains unclear if this specific conformational change is necessary or sufficient for FtsW activation (Fig. S22G). In the FtsQLBWIN simulation, we also observed that FtsW[K370] flipped to form a salt bridge with FtsW[D297] within the first 200 ns that persisted to the end of the simulation (Fig. 5D, Movie S5). This was also observed in FtsI[R167S] and FtsW[E289G] SF simulations, but not in the wild-type FtsQLBWI simulation. The consistency between FtsN[E]-induced conformational changes and those that appeared in simulations of SF complexes with FtsI[E289G] or FtsI[R167S], but not the wild-type FtsQLBWI or the DN FtsL[R61E] complex, suggests that FtsN[E] binding in the Hub region has allosteric effects on FtsW and/or FtsI activity in the FtsQLBWI complex.

### Defining long-range interaction paths of FtsQLB complexes

Analyses described above allow us to identify critical local interactions that are modulated by FtsN[E] binding and may contribute to the activation of FtsWI, but do not reveal how local interactions trigger distal conformational changes in the complex. To investigate this, we compared long-range interaction pathways for wild-type and two active (FtsI[R167S] or FtsN[E]) complexes, as well as for the FtsI[R61E] DN complex. We quantified correlations between pairs of residues using a dynamical network model[56]. Highly correlated residues in a simulation trajectory

of a complex reflect their coordinated motions, allowing us to construct long-range interaction pathways. Because FtsQ is primarily connected to the rest of the complex through its interactions with FtsB in the C-terminal Truss region, and because we wish to identify long-range interactions that affect the activities of FtsW and FtsI, we computed optimal paths through the Truss and Hub regions between FtsQ and putative catalytic residues FtsW[D297,20] and FtsI[S307] (Fig. S24). These paths reveal dominant routes of communication between the proteins. The optimal path ensemble between FtsQ and FtsW[D297] for the DN FtsL[R61E] simulation exhibits greater path density through FtsB, while the other simulation systems exhibit greater path density through FtsL (Figs. 5E and S24A). Interestingly, active FtsI[R167S] and especially FtsN[E] simulations included more FtsI residues in optimal paths connecting FtsI to FtsW, which is consistent with our observation that FtsN[E] binds between FtsL and the FtsI head domain in the Hub region to enhance this interaction. The optimal path ensemble between FtsQ and FtsI[S307] extends through SF residues including FtsB[E56], FtsL[E88], and FtsI[H94], and varied across different complexes (Fig. S24B).

### Comparison with a cryo-EM model of the orthologous complex in *Pseudomonas aeruginosa*

The recent publication of the Cryo-EM structure of the FtsQLBWI complex in *P. aeruginosa*[42] makes it possible to address concerns about the physiological relevance of molecular dynamics of predicted divisome structures in *E. coli*. We found that the global conformations of the *E. coli* FtsQLBWI complex was remarkably similar to the *P. aeruginosa* complex (Fig. S25A). The *P. aeruginosa* cryo-EM structure was reported to exhibit a tilt in its FtsI TPase domain, relative to an AF2-predicted structure. We observed a similar tilt of FtsI early in all simulations and measured an angle of only 3° between the *P. aeruginosa* cryo-EM structure and the *E. coli* FtsQLBWI complex after 1 μs MD. Rapid tilt of the TPase domain towards the membrane was observed within the first 200 ns of MD for simulations of all *E. coli* complexes (Fig. S25B). Furthermore, Käshammer et al. described a change in the interface between the FtsI anchor domain and FtsL, where FtsL-interacting residues in the FtsI anchor domain shift from residues FtsI[203–206] in the AF2 prediction to residues FtsI[208–212] in the cryo-EM structure. We observed a similar conformational change as an interface is formed between FtsI[223–228] (equivalent to *P. aeruginosa* FtsI[209–215]) and FtsL[69–75] upon the addition of FtsN[E] (Fig. S25C). Unlike in the *P. aeruginosa* structure, we did not observe a direct interaction between the FtsI head and anchor domains. Instead, the conformer after 1 μs MD for the *E. coli* complex with FtsN suggests that FtsN bridges the head and anchor domains. This may reflect differing roles and interactions for FtsN in *E. coli* and *P. aeruginosa* divisome.

### Discussion

In this work, we first used single-molecule tracking to determine that *E. coli* FtsQLB forms a complex with both inactive FtsWI on the Z track and active FtsWI on the sPG track (Fig. 1A, B). Building upon this result, we used structure prediction and molecular dynamic simulations to model the structures of FtsQLBWI (Fig. 1C), FtsWI (Fig. 1D), and FtsQLBWIN (Fig. 5A). We then examined the structural models by a set of mutations both experimentally (Figs. 2C–E and 3F) and computationally (Figs. 3C–F, 4C, and 5D–E). Combining these structural models with observations from MD and in biological experiments, we propose a regulatory mechanism in which FtsWI activities depend on FtsQLB and are further activated by FtsN in *E. coli*.

In the absence of FtsQLB, FtsWI adopts flexible conformations with low GTase and TPase activities because FtsWI lacks interactions in the Pivot and Lid regions that stabilize FtsWI conformations required for processive sPG strand polymerization and crosslinking. The addition of FtsQLB stabilizes an extended conformation of FtsI through the Truss region. This conformation supports higher FtsWI activities compared to the background level of FtsWI alone, and could be

sufficient in species that are not dependent on FtsN. The binding of FtsN$^E$ both weakens inhibitory interactions in the CCD interface and enhances activating interactions in the AWI interface of the Hub region, collectively resulting in stronger association of the FtsI head domain and FtsL. These changes promote conformations in which interactions between the FtsI anchor-loop and FtsW ECL4 are modified to open the catalytic pore of FtsW. We propose that this conformational change corresponds to the highest activities of the complex because it remodels the catalytic cavity of FtsW to facilitate processive PG polymerization and to allow the growing PG strand to reach the TPase domain of FtsI.

Further, it remains a possibility that binding of the nascent PG strand to FtsI may be coupled with a transient conformational change to an extended conformation compatible with PG crosslinking as proposed for the RodA-PBP2 elongasome complex[39] and observed in our predicted structures (Fig. S3). This mechanism differs in specific details from those proposed in recent work that also drew from structure prediction[22,42,46]. Despite their differences, each of these hypotheses involves allostery linking periplasmic interactions to distal active sites.

## Roles of FtsQLB in FtsWI activation

The structural models of various complexes allowed us to clarify the seemingly contradictory roles of FtsQLB in activating FtsWI. Using a thioester substrate of FtsI, Boes et al., found that adding purified *E. coli* FtsQ moderately inhibited FtsI TPase activity, while adding FtsL and FtsB had little effect[19]. Marmont et al., found that purified *P. aeruginosa* FtsL and FtsB enhanced FtsW GTase activity as well as FtsQLB[9]. In neither case did the addition of FtsN impact FtsI or FtsW activity. Our model of FtsQLBWI shows that FtsQ has few interactions with members of the complex other than FtsB. Thus, a role of FtsQ is to stabilize the global conformation of the complex via these interactions without directly impacting the activity of FtsW or FtsI. The major role of FtsL and FtsB is to scaffold FtsWI in a conformation poised for activation by clamping both the membrane-distal TPase domain of FtsI and membrane-embedded FtsW. Truncations of β-strands in FtsL$^{Δ6}$, FtsL$^{Δ11}$, FtsL$^{Δ16}$, and FtsI$^{Δ10}$ were not lethal, but resulted in filamentous phenotypes indicating defects in cell division. Stacked β-strands between FtsL, FtsB, and FtsQ are consistent with the observation that C-terminal truncation of FtsL$^{Δ100–121}$ (or FtsL$^{Δ21}$) both abolishes FtsL interaction with FtsQ and also results in a reduced level of full-length FtsB[49]. A mild cell division defect was observed for FtsI$^{Δ10}$ but not for FtsI$^{Δ14}$, which is likely caused by a lower expression level of FtsI$^{Δ10}$ than FtsI$^{Δ14}$ under the same induction levels (Fig. S14). These observations suggest a complex role for C-terminal FtsL-FtsI interaction in the Truss region and are consistent with the observation that deletion of FtsI$^{G571–V577}$ appears to be fully functional[57]. Note that our FtsB single-molecule tracking data indicates that FtsB is associated with FtsWI on both the Z track and the sPG track in *E. coli*, and hence whether an FtsWI complex alone is physiologically relevant is unclear.

## The proposed activation mechanism is consistent with previous studies

The proposed activation mechanism is remarkably consistent with what was previously deduced by genetic studies: following recruitment by FtsQ, the activating signal goes from FtsB to FtsL, then to FtsI, and finally to FtsW[10]. We suggest that the final activation step includes remodeling the central cavity of FtsW to facilitate processive PG polymerization and crosslinking. This step can be achieved by multiple means: SF variants in the Pivot (such as FtsW$^{M269I}$), Hub (such as FtsB$^{E56A}$, FtsL$^{H94Y}$, and FtsI$^{R167S}$), and Lid (such as FtsI$^{K211I}$) regions can correctly position the FtsI anchor domain away from the FtsW pore, or directly remodel the structure defined by the FtsI anchor-loop and FtsW ECL4 to open a channel in FtsW (such as FtsW$^{E289G}$). However, short circuiting this pathway removes potential points of regulation

needed to coordinate cell wall synthesis in space and time and in response to environmental conditions.

This activation mechanism is also consistent with previous mutagenesis studies, as our results provide unprecedented details in potential conformational changes caused by previously identified SF and DN mutants. For example, previously it was observed that substitutions removing the negative charge of FtsW$^{E289}$ do not affect the function of FtsW while only FtsW$^{E289G}$ is superfission[20]. This observation can be explained by the fact that the loss of the side chain of FtsW$^{E289}$ and/or the flexibility of FtsW$^{G289}$, but not necessarily specific interactions, removes the capping of the FtsW cavity and activates FtsW GTase activity. The observation that SF variant FtsW$^{E289G}$ or FtsW$^{M269I}$, but not overexpression of FtsN, can rescue the double-DN mutant FtsL$^{L86F/E87K}$,[10] is also consistent with our proposed activation mechanism and results. Since the double-DN mutant FtsL$^{L86F/E87K}$ loses its ability to bind to FtsN to trigger conformational changes that increase FtsWI activity via allosteric paths extending from the Hub through the Lid and Pivot regions (Fig. 5E), it is expected that Lid and/or Pivot mutations could short-circuit this effect.

The activation mechanism we propose does not specifically involve FtsWI catalytic residues. However, this model could be expanded to address possible regulation of active site conformation and dynamics near catalytic residues such as FtsW$^{D297}$,[20]. For example, dynamics in the vicinity of FtsW$^{D297}$ revealed potential roles of FtsW$^{K370}$ in addition to FtsW$^{Y379}$ in regulating FtsW activity (Fig. 5D). In predictions, FtsW$^{K370}$ blocks a putative substrate channel[39], suggesting that FtsW$^{K370}$ conformation can regulate substrate or product transport. We also did not address in detail the hypothesis of a diprotomeric Fts[QLBWI]$_2$ complex or the role of cytoplasmic FtsL-FtsW interactions[46]. Nevertheless, we observed stable FtsL-FtsW cytoplasmic interaction in our simulations even though the N-terminal cytoplasmic tail of FtsW was truncated. Additionally, the location of the FtsN$^E$ binding site and conformational changes in FtsL and FtsB helices observed in simulations with FtsN$^E$ and FtsI$^{R167S}$ are qualitatively compatible with the diprotomer model that requires flexibility in this region.

Our simulations for all FtsQLBWI wild-type and mutant complexes, with and without FtsN, rapidly attained a bent conformation similar to that observed for the *P. aeruginosa* cryo-EM complex (Fig. S25B). Käshammer et al., hypothesized that a transition from the observed bent conformation to the AF2-predicted straight conformation could be associated with binding of substrate or proteins such as FtsN. Our results here, when further combined with our previous single-molecule imaging results for FtsN[18], demonstrate that FtsN$^E$ is associated with FtsQLBWI on the sPG track, with MD suggesting that FtsN activates the GTase activity of FtsW in the observed bent conformation. We did not observe stable straight conformations in our simulations. Our model also provides insight into conflicting reports about the role of FtsN in different organisms. We observed that a shift in FtsI-FtsL interactions, analogous to that observed between predicted and experimental structures in the *P. aeruginosa* cryo-EM study, only occurred upon the addition of FtsN$^E$ in the *E. coli* complex (Fig. S25C).

This study comes with several limitations. Importantly, the absence of direct measurements of FtsN-FtsQLBWI binding, and simultaneous GTase and TPase activities, restricts our ability to draw definitive conclusions about the roles of individual protein–protein interfaces. Subsequent work can test hypotheses drawn from our findings, leveraging opportunities such as a recently reported *E. coli* divisome complex with in vitro transglycosylase activity[42]. In light of these limitations, we primarily focused on residues at protein–protein interfaces with well-characterized phenotypes. The significance of many newly identified interactions remains to be investigated. Second, while our results show consistency with the *P. aeruginosa* cryo-EM structure, we cannot disregard potential

influences of model-dependence and random effects. Equilibrium MD of 1 μs is insufficient to accurately quantify binding free energies or to confidently identify new interfaces; simulation techniques with enhanced sampling should be considered in future research. Lastly, our simulations exclude FtsW and FtsI substrates as well as divisome proteins known to impact cell-wall synthesis[48]. Filling in these aspects of this picture will enable a more direct correlation between divisome structure and dynamics with substrate binding and enzymatic mechanisms.

In summary, our results are not only consistent with past biochemical and genetic studies, but also shed light on the molecular details of active and inactive conformations of FtsWI. The approach we developed in this work—structure prediction followed by MD simulation with results informing experiments to test hypotheses arising from modeled structures—proved powerful in generating new insights into molecular interactions in the divisome.

## Methods

### Complementation assay

To ensure proper growth of *ΔftsL* and *ΔftsI* strains, arabinose was included during maintenance and preparation to induce expression of FtsL and FtsI, respectively, from the $P_{BAD}$ promoter. Cells were grown overnight at 37 °C in LB and 0.2% arabinose from a single colony. The following day, the saturated culture was reinoculated 1:1000 in fresh LB with 0.2% arabinose. Cells were grown in log phase at 37 °C to an OD600 of 0.5 for all conditions. Cells were then washed three times with 0.9% saline. They were then serial diluted in 0.9% saline and plated on LB plates containing either 0.2% arabinose or 0.4% glucose (to repress expression from the $P_{BAD}$ promoter) and IPTG as noted. Plates were grown at 37 °C overnight and then imaged.

### Plasmid and strain construction

Plasmids used in this study (Table S1) were assembled by Polymerase Chain Reactions (PCR) amplifying insert and vector DNA fragments followed by In-Fusion cloning (Takara Biosciences, In-Fusion HD Cloning Kit). Oligonucleotides (Integrated DNA Technologies) used in PCR amplifications are described in Table S12. All plasmids were verified by DNA sequencing. After construction, electroporation was used to transform plasmids to create strains of interest (Table S2) under appropriate antibiotic selection. Depletion strains were maintained under arabinose induction to maintain wild-type phenotype.

### Single-molecule tracking sample preparation, imaging, and data analysis

Prior to imaging, cells were grown to log phase at 25 °C in defined minimal M9 medium (0.4% glucose, 1x MEM amino acids, and 1× MEM vitamins, M9(+) glucose). Cells were incubated for 20 min with the addition of 50 nM Janelia Fluor 646 (JF646). Following labeling, cells were washed 3 times with M9(−) glucose (0.4% glucose and 1x MEM vitamins). Cells were placed onto a 3% agarose gel pad (M9(−) glucose), sandwiched with a coverslip, and enclosed within an FSC2 chamber (Bioptechs) for imaging. Cells were imaged after 30 min of equilibration on the microscope.

For experiments with fosfomycin, 200 μg/ml of Fosfomycin was added to the gel pad and to the cells following the final wash step. Cells were imaged 1 h after Fosfomycin addition. For experiments with rich, defined medium, EZDRM (Teknova) was used in place of M9 for the growth media, wash buffer, and gel pad.

HaloTag-FtsB tracking was performed on a home-built microscope as previously described[17,30]. Briefly, strains were imaged on an Olympus IX-71 microscope body using an UPLANApo 100XOHR Objective (NA1.50/oil) with a 1.6 x field lens engaged. JF646 was imaged with a 674-nm laser (Coherent) at an excitation power density of ~25 W/cm². The exposure time was 500 ms and the imaging frame rate was 1 Hz.

Single-molecule data analysis was performed as previously described[17,18,30]. Briefly, positions of single molecules were determined using the ImageJ (1.53f51) plug-in, ThunderSTORM (1.3-2014-11-08)[58]. The remaining data analysis and postprocessing was performed using home-built MATLAB scripts available from the Xiao Laboratory GitHub repository (XiaoLabJHU)[17]. In short, using a nearest-neighbor algorithm, single molecules were linked to trajectories. Using the bright-field image as a visual guide, only trajectories near the midplane of a cell's long axis or visible constriction sites were chosen to ensure the measurements were made on single molecules at the cell-division site. The true displacement of tracked molecules around the circumference of a cell is underestimated in 2D single-molecule tracking due to the cylindrical cell envelope. Therefore, the trajectories were unwrapped to one-dimension. Unwrapped trajectories were then segmented manually to determine stationary or processive movement. These procedures were described in detail in[17].

The cumulative probability distribution of directional moving velocities was calculated for each condition and fit to either a single or double log-normal population:

$$CDF = P_1 \frac{1 + erf\left[\frac{lnv - \mu_1}{\sqrt{2\sigma_1}}\right]}{2} + (1 - P_1)\frac{1 + erf\left[\frac{lnv - \mu_1}{\sqrt{2\sigma_2}}\right]}{2}, \quad (1)$$

where $v$ is the moving velocity and $P_1$ is the percentage of the first population. For a single population, $P_1 = 1$. The parameters $\mu$ and $\sigma$ are the natural logarithmic mean and standard deviation of the log-normal distribution. The average velocity was calculated using $\exp\left(\mu + \frac{\sigma^2}{2}\right)$. To estimate errors in parameter estimates, CDF curves were generated for 1000 bootstrapped samples and fit, with the standard deviations of parameter fits estimating 1 standard error of measurement (s.e.m.).

### Bright-field sample preparation, imaging, and western blots

Cells were grown overnight at 37 °C in LB and 0.2% arabinose from a single colony. The following day, the saturated culture was reinoculated 1:1000 in fresh LB with the appropriate inducer and/or repressor (0.2% arabinose, 0.4% glucose, and/or ITPG as noted). Cells were grown to log phase at 37 °C. Cells were then washed three times with 0.9% saline with the appropriate inducer and/or repressor (0.2% arabinose, 0.4% glucose, and/or ITPG as noted). Cells were placed onto a 3% agarose gel pad (0.9% saline with the appropriate inducer) and/or repressor (0.2% arabinose, 0.4% glucose, and/or ITPG as noted), sandwiched with a coverslip, and enclosed within an FSC2 chamber (Bioptechs) for immediate imaging. Phase imaging was performed on a home-built microscope as previously described[30]. Briefly, phase imaging was performed on the same microscope as that used for single-molecule tracking experiment with condenser lamp illumination using a UPLANFLN 100X Objective (NA1.30/oil).

For western blots cells were grown under the same conditions and harvested via centrifugation. Samples were resuspended in 2x Laemmli Sample Buffer (Bio-Rad) and heated for 10 min at 95 °C before loading onto a Mini-PROTEAN TGX Precast Gel (4–15%; Bio-Rad). Electrophoresis, transfer to nitrocellulose and blot development followed standard procedures. The primary antibody was polyclonal rabbit anti-FtsI sera (1:50,000 dilution). The secondary antibody was HRP goat anti-rabbit (1:50,000 dilution; ThermoFisher). Chemiluminescence was detected using a ChemiDoc System (Bio-Rad).

### Prediction of protein complex structures

We used AlphaFold2[23] as implemented in ColabFold[33] using the LocalColabFold version and AlphaFold2 parameters available in September 2021. ColabFold uses the MMseqs2 server[59] to obtain sequence alignments, greatly reducing local storage and computation requirements. Predictions were made on a Google Cloud Platform instance with an A100 GPU, taking approximately 1.5 h including sidechain

relaxation. Template structures were not used, and predictions utilized 48 recycle steps. In preliminary work, we found that AlphaFold2 model 3, which was trained without templates, typically provided the best performance as measured by local pLDDT and global pTMscore metrics. Predictions from model 3 for FtsWI, FtsQLBWI, and FtsQLBWIN were used to generate MD systems. Model 5, which was trained similarly and produced similar predictions for γ-proteobacterial complexes, was later found to reproduce some additional reported interactions in other species and was used for *B. subtilis* and *S. pneumonia* in Fig. S2B. We did not use AlphaFold-multimer[25] for these predictions, as its initial implementation resulted in structures with steric clashes that could not be directly used for building MD systems. We note that this has since been addressed and that later versions of AlphaFold-multimer give similar predictions of divisome complexes to what we report that typically lack clashes and are appropriate for MD. All structure predictions shown in the manuscript, as well as protein complex structures following 1 μs of MD, are available as supplementary data.

## Conservation of FtsN^E

We constructed sequence logos illustrating FtsN^E conservation following previously reported methods applied to the *E. coli* FtsN cytoplasmic domain[60]. Sequences annotated as FtsN (TIGRFAM code TIGR02223) were downloaded from AnnoTree[61] for *Enterobacteriaceae* (513), *Pasteurellaceae* (113), and *Vibrionaceae* (227). Sequences that were incorrectly annotated based on length or other features were manually removed. Multiple sequences alignments were constructed using MUSCLE[62] and sequence logos for regions corresponding to *E. coli* FtsN[73–95] were obtained using WebLogo 3[63]. Only 10 FtsN sequences were identified for *Pseudomonadaceae* following this method, so we obtained 304 sequences homologous to *P. aeruginosa* FtsN[68–118] in *Pseudomonadaceae* using blastp against the RefSeq Select proteins database, verifying that this also reproduced conservation observed for *Enterobacteriaceae* when using *E. coli* FtsN[58–108] in the same way.

## Equilibrium molecular dynamics simulation of FtsQLBWI and FtsQLBWIN complexes with superfission and dominant negative variants

Simulation systems for molecular dynamics were constructed from AlphaFold models of FtsWI, FtsQLBWI, and FtsQLBWIN including the following subunits and residues: FtsQ (20–276), FtsL (1–121), FtsB (1–103), FtsW (46–414), FtsI (19–588), and FtsN (58–108). Mutations of interest were made using the Mutagenesis Wizard of the PyMOL molecular visualization software.

All systems were embedded in a POPE membrane and solvated (see Table S5 for system dimensions) with 150 mM NaCl in TIP3P water[64] using the CHARMM-GUI Membrane Builder. All systems were electrically neutral. N-termini of FtsQ, FtsW, FtsI, and FtsN were capped with the CHARMM ACE patch, and the C-terminus of FtsN was capped with the CHARMM CT3 patch. Equilibration was performed according to the CHARMM-GUI equilibration protocol involving a series of gradually relaxing sidechain and backbone restraints. After equilibration, 5 ns pre-production simulations were performed prior to simulation on the special-purpose supercomputer Anton 2 at the Pittsburgh Supercomputing Center.

Simulations on Anton 2 were performed using the CHARMM36m forcefield in an NPT ensemble at 310 K, 1 atm, and with a timestep of 2.5 fs. Bond lengths for hydrogen atoms were constrained using the M-SHAKE algorithm[65]. An r-RESPA integrator[66] was used; long-range electrostatics were computed every 6 fs. Long-range electrostatics interactions were calculated using the k-space Gaussian split Ewald method[67]. Trajectories were written at an output frequency of 0.24 ns/frame. Each trajectory was unwrapped using the PBCTools plugin of VMD and aligned to the backbone atoms of FtsW (segid PROD) to facilitate comparison across complexes. Code for analyzing simulations was written using the MDAnalysis python package (1.0.0). Hydrogen bonds between relevant interfaces were computed using the HydrogenBondAnalysis function of MDAnalysis (v 1.0.0) in which a hydrogen bond is defined by a distance cutoff of 3.0 Å and an angle cutoff of 150°.

## Calculation of structure parameters

Solvent-accessible surface area was calculated for all hydrophobic FtsL residues in the region of FtsL interacting with FtsB and the FtsI head and anchor domains (residues 62–86 inclusive of FtsL residues L62, L63, A65, L70, V71, L72, A76, L77, I79, W81, L84, I85, and L86) using the *get_area* command in PyMOL (version 2.5) with parameters *dot_solvent*, *dot_density*, and *solvent_radius* set to 1, 4, and 1.4, respectively. FtsI TPase domain tilt angles were calculated following methods applied to the *P. aeruginosa* FtsQLBWI structure[42]. Trajectories were aligned by minimizing RMSD for FtsW $C_\alpha$ atoms, and the coordinates of $C_\alpha$ for FtsI^S307 were tracked in this reference frame. The tilt angle is defined as the angle between two vectors connecting $C_\alpha$ atoms for FtsW^D297 and FtsI^S307 in the AF2-predicted structure and in frames of the trajectory. The *P. aeruginosa* structure was compared by aligning to *E. coli* FtsW using the default settings for the *align* command in PyMol (version 2.5) and analyzing $C_\alpha$ coordinates for equivalent residues.

## Dynamical network analysis for computing optimal paths

Dynamical network representations of the FtsQBLWI, FtsQLBWI^R167S, FtsQL^R61EBWI, and FtsQLBWIN complexes were generated using the dynetan python package (1.0.1)[68]. In the network, each node is defined as the $C_\alpha$ atom of a residue, and pairs of nodes are considered connected if their heavy atoms are within 4.5 Å of each other for at least 75% of the final 500 ns of the trajectory. The strength of an edge between two nodes is calculated using a correlation coefficient computed from a k-nearest-neighbor-based estimator of mutual information[68]. From these correlations, optimal paths were computed using the Floyd–Warshall algorithm implemented in dynetan (1.0.1) and NetworkX[69]. To illustrate long-range interaction paths, a source node was selected as the first simulated residue of FtsQ (FtsQ^N20) and target nodes were selected as the catalytic residues of FtsW (FtsW^D297) and FtsI (FtsI^S307), resulting in two ensembles of optimal paths.

## Reporting summary

Further information on research design is available in the Nature Portfolio Reporting Summary linked to this article.

# Data availability

Source data are provided with this paper as a Source Data file. The all-atom trajectories data generated in this study have been made available in the Anton 2 database [https://antonweb.psc.edu/trajectories/]. Trajectories of protein atoms and analysis code have been deposited in the Zenodo database under accession code https://doi.org/10.5281/zenodo.8042906 [https://zenodo.org/record/8042906]. The hydrogen-bonding data has been provided as Supplementary Data 1. Initial model coordinates and coordinates of each system after 1 μs of simulation are included as Supplementary Data 2. PDB Data: 5OIZ, 7ONO, 5Z2W, 6PL5, 8BH1. Source data are provided with this paper.

# Code availability

Code for analyzing single-molecule tracking data is available from the Xiao Laboratory GitHub repository (XiaoLabJHU; https://github.com/XiaoLabJHU/SMT_Unwrapping)[17]. The code for analyzing the MD data have been deposited in the Zenodo database under accession code https://doi.org/10.5281/zenodo.8042906 [https://zenodo.org/record/8042906].

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

## Acknowledgements

The authors thank all members of the Hensel, Xiao, and Lau laboratories for helpful discussions and feedback on the manuscript. Z.H. received support for this work from FCT—Fundação para a Ciência e a Tecnologia, I.P., through MOSTMICRO-ITQB R&D Unit (UIDB/04612/2020, UIDP/04612/2020, Z.H.) and LS4FUTURE Associated Laboratory (LA/P/0087/2020, Z.H.), from a joint research agreement with the Okinawa Institute of Science and Technology, and from the Google Cloud Research Credits program with the award GCP20210916. S.F.C. received support from FCT Fundação para a Ciência e a Tecnologia, I.P., through a PhD fellowship (PD/BD/135480/2018, SMC). Work in the Xiao laboratory was supported by NIH F32GM143895 (B.M.B.), NIH T32GM007445 (J.W.M), and R35GM136436 (J.X.). Work in the Lau laboratory was funded by the Johns Hopkins Catalyst Award (to A.Y.L.); NIH T32GM135131 (to R.A.Y.). Anton 2 computer time (MCB130045P) was provided by the Pittsburgh Supercomputing Center (PSC) through NIH grant R01GM116961 (to A.Y.L.); the Anton 2 machine at PSC was generously made available by D.E. Shaw Research. We also used resources provided by Advanced Research Computing at Hopkins (ARCH) at Johns Hopkins University.

## Author contributions

B.M.B., J.X., and Z.H. conceived the experiments. R.A.Y. and A.Y.L. designed the simulation workflow. B.M.B. and J.W.M. constructed plasmids and strains for imaging experiments. B.M.B. performed single molecule and phenotype imaging, genetic experiments, and all imaging analysis. R.A.Y. performed molecular dynamic simulations. R.A.Y. and Z.H. wrote analysis code. B.M.B., R.A.Y., A.Y.L, J.X., and Z.H. analyzed molecular dynamics data. S.C. and Z.H. analyzed sequence conservation. B.M.B., R.A.Y., J.X., and Z.H. wrote the original draft. All authors reviewed and edited the manuscript. B.M.B., J.W.M., J.X., and Z.H. acquired funding.

## Competing interests

The authors declare no competing interests.
