## [Peer Review File · Nature Communications]

Conformational changes in the essential E. coli septal cell wall synthesis complex suggest an activation mechanismReviewer #1 (Remarks to the Author):

Dear Authors,
Dear Editors,

Thank you for inviting me to revise the manuscript by Britton et al.

In this manuscript, the authors used the ColabFold (AlphaFold2) server to predict the structure of the core-complex of the bacterial divisome, a key megacomplex promotor of cell division, and a very promising antimicrobial target. Britton et al. predicted the structure of the core-complex of the *E. coli* divisome (FtsQLBWI) and subcomplexes. The authors interpret the predicted structure(s) and identify extensive interactions in four regions of the periplasmic domains (Lid, Pivot, Hub, and Truss). Later, the authors validate the architectural importance of these regions using molecular dynamics simulations, together with mutagenesis and single-molecule tracking microscopy. The results from these experiments lead to propose a mechanistic model of activation of the cell wall synthase sub-complex, formed by FtsW and FtsI, by the action of the regulatory proteins, FtsQ, FtsL, FtsB, and FtsN.

In the last paragraph (Lines 474-476), the authors reveal that a cryo-EM structure of the FtsQLBWI core-complex has been elucidated by another group (REF 53). This structure, solved by Käshammer et al. (REF 53) at a resolution of 3.7Å, shows the architecture of FtsQLBWI from *Pseudomonas aeruginosa*, which shares an average similarity of about 50% with the *E. coli* FtsQLBWI complex. Unfortunately, the coordinates of this structure have not been released, although they have been deposited (PDB code: 8BH1). Thus, Britton et al. cannot compare the model predicted by AlphaFold2 with the cryo-EM structure by Käshammer et al. (REF 53). The only comparison that can be done at this point is to put side-by-side Figure 1C (this manuscript), and Figure 1E from REF53, which shows the cryo-EM structure. I was surprised to observe evident differences between the FtsQLBWI complex predicted by AlphaFold2 and the cryo-EM structure. For example, the conformations of all the periplasmic domains (FtsI, L, B, and Q), relative to the position of FtsW are strikingly different. In addition, folded regions observed in the AlphaFold2 prediction, are 1) not folded in the cryo-EM structure, 2) are pointing in different directions, or 3) look entirely different.

Given that the ultimate experimental validation of the model predicted by AlphaFold2 would be a cryo-EM structure of the FtsQLBWI core-complex, and that MD simulations are as reliable as the model that is used to run calculations, I find the manuscript by Britton et al. lacking strong high-resolution experimental evidence that supports their mechanistic claims. I understand that the authors put a lot of effort into validating this predicted model using single-molecule tracking microscopy and mutagenesis analysis in live cells. However, these methods allow imaging at only very low resolution (not comparable to single particle cryo-EM methods) and can lead to misinterpretation, e.g. cell division defects can be caused by unrelated impaired cell processes, not necessarily by destabilisation of the identified interaction domains (Lid, Pivot, Hub, and Truss). For a reader unfamiliar with structural biology methods, this manuscript would seem to present strong mechanistic insight into how the FtsQLBWI core-complex works, but unfortunately, this is not the case.

Taking into consideration 1) the alarming differences between the model predicted by AlphaFold2 and the cryo-EM structure (REF 53), and 2) the lack of high-resolution methods behind the validation of the predicted model (e.g high-resolution structures, crosslinking analysis, biophysical assays with purified material, etc), I cannot recommend the publication of this manuscript in Nature communications, a journal that is known for publishing strong experimental evidence supporting mechanistic models.

I would suggest that the authors wait for the coordinates of the cryo-EM structure to be released, or solve themselves cryo-EM structures of the core-complex and subcomplexes, and then together with MD simulations, single molecule tracking, assays with live cells, and other biophysical methods, revise and complete the experiments and conclusions presented here.

Reviewer #2 (Remarks to the Author):

Britton and coauthors use an interdisciplinary combining structure prediction, atomistic modelling, single-molecule imaging, and mutagenesis (both in silico and in vivo) to study protein-protein interactions formed by different membrane proteins: FtsW, FtsI, FtsQ, FtsL, FtsB, and FtsN. More specifically, in term of modelling, they have used Alpha Fold predictions combined with MD simulations to try to explain how the complex FtsWI can be regulated by other proteins FtsQ, FtsL, FtsB, and FtsN. They found that FtsQLB seemed to form a scaffold to maintain FtsWI in an upward position and removing the FtsQLB proteins resulted in a collapse of the FtsWI on the membrane. In function of the partner protein, Britton and coauthors identified changes at the different interfaces as well as conformational changes that the authors linked to different levels of activations/inactivations.

It is an interesting study highlighting the importance of interactions of regulators of FtsWI to switch between different signaling states. This reviewer have thought several concerns that need to addressed. Especially, the modelling is based on an Alpha Fold prediction and the use of MD simulations seemed quite limited to discriminate between genuine and artefactual results.

Major points:

1- Authors embedded the protein into a membrane but then often hide this membrane which limits the understanding of the positioning of the transmembrane regions. It is especially striking on Movie S5 where the transmembrane helix of FtsQ seems to largely go outside the membrane position schematized by two black lines. So, it would be useful to at least display the position of the lipid polar groups to delineate this membrane. I also strongly encourage the authors to release coordinate files including the protein embedded into the membrane and not only the coordinates of the protein.

2- Analyses of C-alpha seems a very simple measurement to get insights into protein-protein interactions. I would recommend going a bit further and really assess protein-protein interactions for all the residues at the different interfaces using distance matrices and h-bond analyses. This would also help the readers to really get a clear idea of the overall stability of the Alpha Fold model.

3- for the collapsing of the protein, how much reproducible this collapse is ? Can you repeat this simulations 2-3 times to see if it is the same final structure or if there is a degree of flexibility ? Is there some interactions with the underlying lipids. How this collapse can be validated experimentally speaking ?

4- Overall, I am not sure that 1 μ s simulation is enough to really assess the stability of an Alpha Fold model. Simulations using Anton supercomputer allowed simulating long timescale simulations (up to hundreds of μ s see eg <https://rupress.org/jgp/article/155/2/e202213085/213765/Gating-and-modulation-of-an-inward-rectifier>), so simulating, at least for the FtsQLBWI, up to 5 μ s would be useful to better assess the model stability.

Reviewer #3 (Remarks to the Author):

In Britton and Yovanno et al., the authors explore the molecular mechanism by which E. coli FtsQLB and FtsN activate PG synthesis by the FtsW-FtsI enzymatic complex using AlphaFold-based structural predictions, in vivo genetic and imaging analyses and molecular dynamics (MD) simulations. Specifically, they attempt to elucidate the following mechanistic points: a) the hierarchical order of activation by FtsQLBN components, b) the role of FtsQLB and FtsN – activating or inhibitory – in this process, and c) the conformational rearrangements that lead to enzymatic activation of FtsWI.

To establish the order of activation, the authors carry out single-particle tracking experiments in cells, using methodology previously reported by the Xiao group, and show that FtsB has both slow-

and fast-tracking populations i.e., that it is associated with both actively synthesizing and inactive complexes, respectively. Taken together with an earlier result that FtsN exhibits only slow-moving dynamics, this experiment suggests that regulation by FtsN occurs after that by FtsQLB, in the later stages of divisome assembly/activation. To resolve the regulatory roles of FtsQLB vs FtsN, the authors use AlphaFold-generated models to visualize the putative interaction interfaces between all the components and carry out MD simulations to evaluate the relative stabilities of different sub-complexes. They observe that the extended conformation of FtsI (presumably, an activated state) is destabilized when FtsWI is not bound by FtsQLB, suggesting that these components promote structural opening of FtsI. To probe the contributions of different interfaces to regulation, the authors carry out MD simulations, comparing WT dynamics with those of previously reported dominant-negative and superfission mutants in each region. In two instances computational analysis is complemented with experiments: the authors test the effect of several truncations in the Truss region on division (Fig. 2) and quantify the cellular dynamics of a superfission mutant in the Hub/AWI region (Fig. 3). From these data the authors conclude that FtsQLB partially activates FtsWI, leaving it in an autoinhibited state that requires further activation by FtsN. FtsN, in their model, accomplishes the final regulatory step, relieving inhibitory contacts and stabilizing the activated conformation of FtsWI.

While this model seems reasonable, it does not explain earlier work albeit in a different organism showing that the presence of FtsQLB dramatically impacts polymerization activity of FtsWI, while further addition of FtsN has no effect (Marmont LS, Bernhardt, TG PNAS 2020). In summary, my main concern is that the AlphaFold models and simulations generated in this study are not sufficiently supported by experimental evidence. As a result, the validation of the proposed model largely relies on insights from earlier genetic work, as well as structural and mechanistic frameworks presented in recent pre-prints.

Major concerns:

1. The AlphaFold-generated structural predictions presented in the study and the proposed mechanistic models are not sufficiently supported by experimental or computational evidence.

Notably, several earlier preprints explore a similar set of questions using AlphaFold (from the Senes group <https://doi.org/10.1101/2022.10.30.514410>), cryo-EM (from the Löwe group, <https://doi.org/10.1101/2022.11.21.517367>), and in vitro single-molecule imaging (from the Bernhardt, Loparo and Kruse groups, <https://doi.org/10.1101/2022.11.07.515454>). The cryo-EM structure of the FtsQLBWI complex from *P. aeruginosa* defines key interaction interfaces between all the components (with the exception of FtsN) and proposes a mechanism of activation, whereby contacts between FtsI and FtsQLB alter the conformation of FtsI. The single-molecule study establishes the mechanism of allosteric activation in the evolutionarily-related PG synthase from the Rod complex (RodA-PBP2), showing that structural rearrangement of PBP2 into an extended state serves as the ON-switch for enzymatic activation.

The authors are asked to comment on how their findings fit into the context of these – and other – studies in the field, and to highlight novel mechanistic insights of their manuscript.

2. Truncation analysis that probes the role of the beta-sheet sandwich in the stabilization of the FtsQLBI complex (Fig. 2) lacks key control experiments. For instance, the authors do not quantify expression levels of truncated constructs (e.g., via westerns) and do not include microscopy analysis showing cellular localization of truncated FtsI variants. In the absence of these data, it is unclear whether division activation/defects arise from differences in the cellular expression or localization of different variants or changes in binding/activity. It is also hard to interpret the seemingly self-contradictory results that truncation of the beta-sheet region for FtsL results in mislocalization of FtsL and cell division defects, while FtsI beta-sheet truncations promote division.

3. For some mechanistic claims derived from AF models, there are no in vitro or in vivo experiments to complement MD simulations and to help validate the proposed mechanistic claims (Fig. 3-5). In general, it is hard to interpret MD simulations in the absence of mutational analysis coupled with either binding assays or co-localization imaging that can probe proposed changes in binding affinity directly. The MD simulation analysis itself is fairly limited, and it is not clear how some of the observed conformational changes contribute to binding/activity (as outlined in

examples below).

Fig. 3: To what extent does a 1 Å shift in the packing between FtsL and FtsI at the AWI interface contribute to the overall binding energy?

Fig. 4: How much do local changes in the conformation of the FtsI anchor loop change substrate accessibility in the cavity of FtsW?

Minor comments:

1. The figures are hard to navigate: the authors could consider simplifying the cartoons and annotating them more explicitly, as well as using distinct colors for proteins that form close interfaces.

Point-to-point Responses to Reviewers comments for Nature Communications manuscript NCOMMS-22-53130-T: Conformational changes in the essential *E. coli* septal cell wall synthesis complex suggest an activation mechanism

For clarity, we *italicized reviewers' comments* and colored our responses in blue.

Reviewer #1 (Remarks to the Author):

Dear Authors,

Dear Editors,

Thank you for inviting me to revise the manuscript by Britton et al.

*In this manuscript, the authors used the ColabFold (AlphaFold2) server to predict the structure of the core-complex of the bacterial divisome, a key megacomplex promotor of cell division, and a very promising antimicrobial target. Britton et al. predicted the structure of the core-complex of the *E. coli* divisome (FtsQLBWI) and subcomplexes. The authors interpret the predicted structure(s) and identify extensive interactions in four regions of the periplasmic domains (Lid, Pivot, Hub, and Truss). Later, the authors validate the architectural importance of these regions using molecular dynamics simulations, together with mutagenesis and single-molecule tracking microscopy. The results from these experiments lead to propose a mechanistic model of activation of the cell wall synthase sub-complex, formed by FtsW and FtsI, by the action of the regulatory proteins, FtsQ, FtsL, FtsB, and FtsN.*

We thank the reviewer for their concise summary and wish to clarify that we analyzed interfaces in coordinates obtained after 1- \$\mu\$ s molecular dynamics simulations. We did not interpret the AlphaFold2 (AF2)-predicated structures directly. In **Supplementary Fig. 3**, we compared the structures before and after MD and showed that there are large conformational changes in the transpeptidase (TPase) domain of FtsI.

In our revision, we showed the dynamics of this conformational change by plotting TPase domain tilt in **Supplementary Fig. 25B**.

*In the last paragraph (Lines 474-476), the authors reveal that a cryo-EM structure of the FtsQLBWI core-complex has been elucidated by another group (REF 53). This structure, solved by Käshammer et al. (REF 53) at a resolution of 3.7Å, shows the architecture of FtsQLBWI from *Pseudomonas aeruginosa*, which shares an average similarity of about 50% with the *E.coli* FtsQLBWI complex. Unfortunately, the coordinates of this structure have not been released, although they have been deposited (PDB code: 8BH1). Thus, Britton et al. cannot compare the model predicted by AlphaFold2 with the cryo-EM structure by Käshammer et al. (REF 53). The only comparison that can be done at this point is to put side-by-side Figure 1C (this manuscript), and Figure 1E from REF53, which shows the cryo-EM structure. I was surprised to observe evident differences between the FtsQLBWI complex predicted by AlphaFold2 and the cryo-EM structure. For example, the conformations of all the periplasmic domains (FtsI, L, B, and Q), relative to the position of FtsW are strikingly different. In addition, folded regions observed in the AlphaFold2 prediction, are 1) not folded in the cryo-EM structure, 2) are pointing in different directions, or 3) look entirely different.*

Given that the ultimate experimental validation of the model predicted by AlphaFold2 would be a cryo-EM structure of the FtsQLBWI core-complex, and that MD simulations are as reliable as the

model that is used to run calculations, I find the manuscript by Britton et al. lacking strong high-resolution experimental evidence that supports their mechanistic claims. I understand that the authors put a lot of effort into validating this predicted model using single-molecule tracking microscopy and mutagenesis analysis in live cells. However, these methods allow imaging at only very low resolution (not comparable to single particle cryo-EM methods) and can lead to misinterpretation, e.g. cell division defects can be caused by unrelated impaired cell processes, not necessarily by destabilisation of the identified interaction domains (Lid, Pivot, Hub, and Truss). For a reader unfamiliar with structural biology methods, this manuscript would seem to present strong mechanistic insight into how the FtsQLBWI core-complex works, but unfortunately, this is not the case.

Taking into consideration 1) the alarming differences between the model predicted by AlphaFold2 and the cryo-EM structure (REF 53), and 2) the lack of high-resolution methods behind the validation of the predicted model (e.g high-resolution structures, crosslinking analysis, biophysical assays with purified material, etc), I cannot recommend the publication of this manuscript in *Nature communications*, a journal that is known for publishing strong experimental evidence supporting mechanistic models.

I would suggest that the authors wait for the coordinates of the cryo-EM structure to be released, or solve themselves cryo-EM structures of the core-complex and subcomplexes, and then together with MD simulations, single molecule tracking, assays with live cells, and other biophysical methods, revise and complete the experiments and conclusions presented here.

Thanks to Dr. Jan Löwe's collegiality, we obtained the cryo-EM structure of the *P. aeruginosa* FtsQLBWI complex in Käshammer et al., *BioRxiv*, 2022. In recent days this structure has been released from embargo in the PDB. The PDB entry notes that the structure will be reported in *Nature Microbiology*. As shown below, the side-to-side comparison (**Response Fig. 1A**) of the *P. aeruginosa* FtsQLBWI complex (left), *E. coli* FtsQLBWI (middle) and *E. coli* FtsQLBWIN (in complex with the essential domain of FtsN, right), and the corresponding contact maps (**Response Fig. 1B**) depict a largely similar organizations and interactions of the five proteins in the complex.

In our revision, **Supplementary Fig. 25A** conveys this information and both conformers after MD and coordinates for full trajectories will be made available to allow for comparisons.

In revision, we added a section (lines 415 to 434) and **Supplementary Fig. 25** comparing the cryo-EM model of the *P. aeruginosa* divisome to conformers for *E. coli* FtsQLBWI and FtsQLBWIN after MD that we argue provides insight into the role of FtsN in *E. coli*.

Specifically to address the concern of the physiological relevance of the AF2 prediction of *E. coli* FtsQLBWI, we compared conformational changes described in Käshammer et al. between predicted and experimental structures to evolution of *E. coli* FtsQLBWI from the AF2 prediction during MD. Käshammer et al. focus on two changes between AF2-predicted and observed *P. aeruginosa* divisome structures. They report a tilt of $\sim 30^\circ$ measured by the angle formed between FtsI active-sites in predicted and experimental structures and the FtsW active site after alignment to FtsW. We observe a similar tilt early in all simulations and measured an angle of 3° between *P. aeruginosa* Cryo-EM FtsQLBWI and *E. coli* FtsQLBWI after 1- μ s MD simulation (**Supplementary Fig. 25A** and **25B**). Käshammer et al. also describe a change in the interface between the FtsI anchor domain and FtsL, where FtsL-interacting residues in FtsI's anchor domain shift from residues 203–206 to residues 208–212. We observed a similar conformational change in which

an interface is formed between FtsI 223–228 (equivalent to *P. aeruginosa* FtsI ~209–215) and FtsL 69–75, but only upon the addition of FtsN^E to the complex (**Response Figure 1B**, arrows, and **Supplementary Fig. 25C**). The change upon FtsN addition may reflect differing roles and interactions for FtsN in *E. coli* and *P. aeruginosa* divisome.

Response Fig. 1. (A) Side-to-side comparison of the cryo-EM structure of *P. aeruginosa* FtsQLBWI (left), *E. coli* FtsQLBWI in the last frame of the MD simulation (middle) and *E. coli* FtsQLBWI^N in complex with the essential domain of FtsN^E in the last frame of the MD simulation (right). **(B)** Contact map of each complex using a cut off value of 6-Å heavy-atom separation between a pair of residues. From top to bottom and from left to right the order is FtsQ, FtsL, FtsB, FtsW and FtsI. In the *E. coli* FtsQLBWI^N contact map, FtsN^E is not shown. Regions absent from the *Pa* model are not excluded from the *Ec* complex contact maps to facilitate comparison.

We next note differences between our data and data the *P. aeruginosa* FtsQLBWI cryo-EM structure:

- (1) The POTRA and transmembrane (TM) domains of FtsQ (gray) and the N-terminal section of the extracellular loop 4 (ECL4) of FtsW (orange) are not included in the cryo-EM structural

model due to missing density in cryo-EM data. The authors suggest that unresolved regions are flexible, and show that the FtsQ POTRA domain likely tilts down towards the membrane. This conformation differs from a published crystal structure (Käshammer *et al* Figure S2D) but resembles our observations. Our MD simulations confirm that regions unresolved by cryo-EM are flexible on the 1- μ s timescale (**Supplementary Movie S3**).

- (2) The orientation of the FtsI head and anchor-loop regions of the pedestal domain show the most significant differences between the *P. aeruginosa* cryo-EM and the conformations of our *E. coli* models following MD. We note that the addition of FtsN^E stabilizes the FtsI head and anchor domains in conformations more closely resembling the *P. aeruginosa* complex.
- (3) FtsN is not included in the *P. aeruginosa* cryo-EM structure as Käshammer *et al.* reported that it was not possible to purify stable *E. coli* FtsQLBWIN complexes. We report modeled FtsN interactions with the FtsI head domain and in the hub region linking FtsI and FtsL. The modeled interface includes residues with known cell-division phenotypes for FtsI, FtsL, and FtsN. We further increased our confidence in the interface through analysis of sequence conservation and by finding similar interface predictions for FtsQLBN (in the absence of FtsWI) and FtsWIN (in the absence of FtsQLB) complexes (**Supplementary Fig. S22**).

In the revision, we placed our proposed activation mechanism in the context of these comparisons and discuss potential limitations and further experiments (lines 520 to 544).

Reviewer #2 (Remarks to the Author):

Britton and coauthors use an interdisciplinary combining structure prediction, atomistic modelling, single-molecule imaging, and mutagenesis (both in silico and in vivo) to study protein-protein interactions formed by different membrane proteins: FtsW, FtsI, FtsQ, FtsL, FtsB, and FtsN. More specifically, in term of modelling, they have used Alpha Fold predictions combined with MD simulations to try to explain how the complex FtsWI can be regulated by other proteins FtsQ, FtsL, FtsB, and FtsN. They found that FtsQLB seemed to form a scaffold to maintain FtsWI in an upward position and removing the FtsQLB proteins resulted in a collapse of the FtsWI on the membrane. In function of the partner protein, Britton and coauthors identified changes at the different interfaces as well as conformational changes that the authors linked to different levels of activations/inactivations.

It is an interesting study highlighting the importance of interactions of regulators of FtsWI to switch between different signaling states. This reviewer has thought several concerns that need to be addressed. Especially, the modelling is based on an Alpha Fold prediction and the use of MD simulations seemed quite limited to discriminate between genuine and artefactual results.

We agree with the reviewer that interpretation of the structural model based on AF2 prediction and the use of MD simulations

Major points:

1- Authors embedded the protein into a membrane but then often hide this membrane which limits the understanding of the positioning of the transmembrane regions. It is especially striking on Movie S5 where the transmembrane helix of FtsQ seems to largely go outside the membrane position schematized by two black lines. So, it would be useful to at least display the position of the lipid polar groups to delineate this membrane. I also strongly encourage the authors to release coordinate files including the protein embedded into the membrane and not only the coordinates of the protein.

We note that the bilayer is not constrained to be flat in our simulations (only needs to be continuous across periodic boundaries), so the abstract representation in **Supplementary Movies S3-S5** is not always accurate.

In the revision, we have added coordinates for all final conformers including membrane atoms. In addition, all-atom trajectories are available in the Anton data repository following publication. We have modified **Movie S4** to include the lipid polar groups.

2- Analyses of C-alpha seems a very simple measurement to get insights into protein-protein interactions. I would recommend going a bit further and really assess protein-protein interactions for all the residues at the different interfaces using distance matrices and h-bond analyses. This would also help the readers to really get a clear idea of the overall stability of the Alpha Fold model.

In **Fig. 1F**, we compare the distribution of $C\alpha$ - $C\alpha$ distances for FtsWI and FtsQLBWI in order to illustrate an impact of global conformational change on the separation between hydrophobic residues that were previously identified as significant. In other scenarios, we used side chain center-of-geometry to quantify hydrophobic packing, and hydrogen bond frequency to quantify hydrogen bonding interactions (e.g. **Fig. 3A, 4A, 4B**).

In revision, we have clarified descriptions of these order parameters to emphasize that they are used to follow the dynamics of conformational change for interfaces with specific hydrophobic and hydrogen bonding interactions.

Regarding a more holistic view of protein-protein interactions, we found that an exhaustive global analysis of interactions was unwieldy. We opted to focus on observational analysis of the dynamics of interactions involving residues previously identified to have major cell-division phenotypes. We have investigated contact maps (see response to reviewer 1) and distance matrices as options to represent data and found that order parameters such as TPase domain tilt (**Supplemental Fig. 25B**) communicated conformational change more effectively.

Regarding global H-bond analysis, our revision includes a supplementary data file showing trajectories of all hydrogen bonding interactions between complex subunits that occur in 50 ns or more of the final 500 ns of MD for all simulations. This makes it possible to examine hydrogen bonds that we did not highlight and, for example, to look at the H-bonding trajectory for FtsI^{Q266}-FtsW^{R60} that we show in **Fig. 1F** and describe by $C\alpha$ - $C\alpha$ distance in **Fig. S6**.

3- for the collapsing of the protein, how much reproducible this collapse is? Can you repeat this simulations 2-3 times to see if it is the same final structure or if there is a degree of flexibility? Is there some interactions with the underlying lipids. How this collapse can be validated experimentally speaking?

The collapse of FtsI onto the membrane in the FtsWI simulation occurs rapidly within approximately 200 ns (**Supplementary Movie S4**). FtsI then remains in the collapsed state for the remainder of the simulation. Although we do not know the extent to which the collapse observed in one FtsWI simulation is reproducible, we note that tilting of FtsI in all simulations indicates that extended FtsWI structures predicted by AlphaFold2 and AlphaFold-Multimer are unstable. As noted in response to Reviewer 1, this observation is consistent with *P. aeruginosa* cryo-EM data.

In the revision, we show the dynamics of FtsI TPase domain tilt for all simulations and observe a similarly rapid, but smaller tilt towards the membrane for other simulation (**Supplementary Fig. S25B**). This is also consistent with previous data showing that FtsQLB plays an essential role in supporting the activity of FtsWI, and with data newly reported in our manuscript showing that FtsB is part of the active sPG synthesis complex. Cryo-EM data for the FtsWI paralog RodA-PBP2 (Sjodt et al., Nat. Micro, 2020) showed diverse collapsed conformations. Thus, since we do not predict that FtsI in the absence of FtsQLB has a unique collapsed conformation, we focused on simulations of FtsQLBWI variants and FtsQLBWIN rather than exploring FtsWI further.

Regarding the potential interaction of FtsWI with the membrane, it is an interesting possibility that that FtsI may interact specifically with lipid head groups. However, due to the focused scope of the current work, we do not plan to investigate this question further.

4- Overall, I am not sure that 1 μ s simulation is enough to really assess the stability of an Alpha Fold model. Simulations using Anton supercomputer allowed simulating long timescale simulations (up to hundreds of μ s see eg <https://rupress.org/jgp/article/155/2/e202213085/213765/Gating-and-modulation-of-an-inward-rectifier>), so simulating, at least for the FtsQLBWI, up to 5 μ s would be useful to better assess the model stability.

We agree that 1- μ s MD is insufficient to assess stability with respect to unbinding or unfolding. For this reason, we were careful not to make claims implying knowledge of the full landscape of diverse conformations in equilibrium. We note that 100 μ s may still not be sufficient to address uncertainty in binding stability. Importantly, without a specific hypothesis to test, it is unclear whether additional simulation would reveal new information not captured in the 1- μ s simulation. Lastly, we currently do not have any allocation on Anton for this project.

Previous studies have shown that most sidechain fluctuations responsible for forming interfaces occur on nanosecond timescales (<https://doi.org/10.1063/1.4934504>). The hydrogen-bonding analysis (**Supplemental Figs S16, 18, 19, and 20**) and the corresponding time trajectories demonstrate that hydrogen bonds break and re-form during 1- μ s simulations. These results suggest some plasticity in binding interfaces consistent with a complex that undergoes conformational changes. Importantly, we were relieved to see that major conformational changes we observed in all FtsQLBWI simulations are reflected in experimental cryo-EM data as indicated in our response to Reviewer 1.

In the revision, we have discussed limitations of the MD approach and the possibility of further experiments with enhanced sampling methods to investigate facets of the system that are not accessible by all-atom MD at equilibrium.

Reviewer #3 (Remarks to the Author):

In Britton and Yovanno et al., the authors explore the molecular mechanism by which E. coli FtsQLB and FtsN activate PG synthesis by the FtsW-FtsI enzymatic complex using AlphaFold-based structural predictions, in vivo genetic and imaging analyses and molecular dynamics (MD) simulations. Specifically, they attempt to elucidate the following mechanistic points: a) the hierarchical order of activation by FtsQLBN components, b) the role of FtsQLB and FtsN – activating or inhibitory – in this process, and c) the conformational rearrangements that lead to enzymatic activation of FtsWI.

To establish the order of activation, the authors carry out single-particle tracking experiments in cells, using methodology previously reported by the Xiao group, and show that FtsB has both slow- and fast-tracking populations i.e., that it is associated with both actively synthesizing and inactive complexes, respectively. Taken together with an earlier result that FtsN exhibits only slow-moving dynamics, this experiment suggests that regulation by FtsN occurs after that by FtsQLB, in the later stages of divisome assembly/activation. To resolve the regulatory roles of FtsQLB vs FtsN, the authors use AlphaFold-generated models to visualize the putative interaction interfaces between all the components and carry out MD simulations to evaluate the relative stabilities of different sub-complexes. They observe that the extended conformation of FtsI (presumably, an activated state) is destabilized when FtsWI is not bound by FtsQLB, suggesting that these components promote structural opening of FtsI. To probe the contributions of different interfaces to regulation, the authors carry out MD simulations, comparing WT dynamics with those of previously reported dominant-negative and superfission mutants in each region. In two instances computational analysis is complemented with experiments: the authors test the effect of several truncations in the Truss region on division (Fig. 2) and quantify the cellular dynamics of a superfission mutant in the Hub/AWI region (Fig. 3). From these data the authors conclude that FtsQLB partially activates FtsWI, leaving it in an autoinhibited state that requires further activation by FtsN. FtsN, in their model, accomplishes the final regulatory step, relieving inhibitory contacts and stabilizing the activated conformation of FtsWI.

While this model seems reasonable, it does not explain earlier work albeit in a different organism showing that the presence of FtsQLB dramatically impacts polymerization activity of FtsWI, while further addition of FtsN has no effect (Marmont LS, Bernhardt, TG PNAS 2020). In summary, my main concern is that the AlphaFold models and simulations generated in this study are not sufficiently supported by experimental evidence. As a result, the validation of the proposed model largely relies on insights from earlier genetic work, as well as structural and mechanistic frameworks presented in recent pre-prints.

Our *E. coli* FtsQLBWI model is indeed consistent with what was observed in Marmont *et al.*, PNAS 2020 in which the presence of FtsQLB dramatically enhances the polymerization activity of FtsWI in the *P. aeruginosa* FtsQLBWI complex. We also note that K ashammer *et al.* reported comparable FtsWI activity for *P. aeruginosa* and *E. coli* FtsQLBWI complexes (the latter including a chimeric FtsWI). This enhancement can be explained by our observation that FtsQLB scaffolds FtsWI in a stable, extended conformation.

It is interesting that the addition of FtsN has no further effect to enhance the *P. aeruginosa* FtsQLBWI complex. As we discussed above in response to Reviewer 1, the *P. aeruginosa* FtsQLBWI cryo-EM and our *E. coli* FtsQLBWI models differ in FtsI head domain orientation and FtsI anchor-loop conformation. The addition of FtsN^E stabilizes the FtsI head domain in a conformation more closely reflecting that observed in *P. aeruginosa* FtsQLBWI. We argue that differences in sequence and structure for FtsI head domains and FtsN residues N-terminal of those binding the hub region suggest why FtsN plays a somewhat different role in *E. coli* than in *P. aeruginosa*. FtsN may play a more essential role in the *E. coli* complex to mediate conformational changes of the FtsI head and anchor domains.

We feel that this discussion is overly speculative without additional investigation, so we focused on FtsN interactions in the hub region and correlated conformational changes. However, reviewers may be interested in speculation on this point. We suggest that differences in FtsI head domain and FtsN sequences may reflect divergence in interactions between the FtsI head domain,

the FtsI anchor loop, and FtsN. The *P. aeruginosa* cryo-EM model shows the head domain directly interacting with the anchor loop, while in *E. coli* this interaction may be mediated by FtsN (**Response Fig. 2**). It is intriguing that, towards the end of 1 μ s MD, FtsN^E interacts with FtsI in a region where mild dominant negative mutations impacting FtsN localization (Wissel *et al.*, *J. Bact.*, 2004). However, we opted to only briefly mention this observation (**Supplementary Fig. 25C**) as it requires further investigation and involves residues outside of FtsN^E.

In the revision, we have clarified our analysis on this point and especially our discussion of the impact of FtsN with respect to observations for the *P. aeruginosa* complex.

Response Fig. 2: Left: In the *P. aeruginosa* cryo-EM model, the FtsI anchor domain is directly interacting with the head domain. Right: In the last frame of MD simulation of *E. coli* FtsQLBWIN, FtsN mediates interaction between the head and anchor domain via a residue (FtsI R63) adjacent to previously identified mild dominant negative residues with FtsN-recruitment defects (FtsI G57, S61, L62, R210).

Major concerns:

1. The AlphaFold-generated structural predictions presented in the study and the proposed mechanistic models are not sufficiently supported by experimental or computational evidence.

Notably, several earlier preprints explore a similar set of questions using AlphaFold (from the Senes group <https://doi.org/10.1101/2022.10.30.514410>), cryo-EM (from the Löwe group, <https://doi.org/10.1101/2022.11.21.517367>), and in vitro single-molecule imaging (from the Bernhardt, Loparo and Kruse groups, <https://doi.org/10.1101/2022.11.07.515454>). The cryo-EM structure of the FtsQLBWI complex from *P. aeruginosa* defines key interaction interfaces between all the components (with the exception of FtsN) and proposes a mechanism of activation, whereby contacts between FtsI and FtsQLB alter the conformation of FtsI. The single-molecule study establishes the mechanism of allosteric activation in the evolutionarily-related PG synthase from the Rod complex (RodA-PBP2), showing that structural rearrangement of PBP2 into an extended state serves as the ON-switch for enzymatic activation.

The authors are asked to comment on how their findings fit into the context of these – and other – studies in the field, and to highlight novel mechanistic insights of their manuscript.

The Senes group (Craven *et al.*, *BioRxiv*, 2022) used AlphaFold2 to predict the structure of the *E. coli* FtsQLBWI complex, finding results approximately equivalent to what we utilized to build MD systems. The authors explored how the predicted complex could be reconfigured to form a Fts[QLBWI]₂ diprotomeric complex, which we did not address as we do not have a model for MD or relevant experimental data to add. The Senes group also characterized mutations impacting cytoplasmic and transmembrane interactions between FtsL and FtsW, which complement our study as we focused on the periplasmic interactions among the complex.

For the comparison with the *P. aeruginosa* FtsQLBWI cryo-EM model, please see the response to Reviewer 1. Briefly, we observed that (1) the protein-protein interfaces in complexes are largely similar, (2) global conformational change (FtsI tilting relative to FtsW, **Supplementary Fig. 25A** and **25B**) that happens consistently in MD closely resembles that observed in *P. aeruginosa* cryo-EM compared to structure prediction, and (3) differences in FtsI head domain and anchor-loop conformations suggest divergence in the role of FtsN. As the reviewer pointed out, the cryo-EM structure study suggests that “contacts between FtsI and FtsQLB alter the conformation of FtsI” and we measured a similar altered conformation to that observed by cryo-EM.

As for comparison of the *E. coli* FtsQLBWI complex with the *T. thermophilus* RodA-PBP2 complex (Sjodt *et al.*, *Nat Micro*, 2020), and the single-molecule FRET work (Shlosman *et al.*, *BioRxiv*, 2022), we note that there are two major differences. First, the periplasmic domain of FtsI in our model rotates to the opposite direction of PBP2 in the RodA-PBP2 crystal structure (**Supplementary Fig. S5C**). This difference was also noted in the cryo-EM structure by Käshammer *et al.* Second, interaction between *T. thermophilus* RodA and PBP2 is mediated by a loop in PBP2 that is significantly shorter in *E. coli* FtsI (**Response Fig. 3**). Together, the relevance of observations in RodA-PBP2 to regulation of FtsQLBWI is unclear. Discussion of mechanisms involving conformational change between compact and extended FtsI conformations with analogy to RodA-PBP2 data can be found in recent work by the Senes and Lowe groups (Craven *et al.*, *BioRxiv*, 2022; Käshammer *et al.*, *BioRxiv*, 2022). We did not find anything in our data to add to this discussion.

In the revision, we added these points in the main text and clarified references to RodA-PBP2 (lines 206 to 212, 415 to 434 and 520 to 544).

Response Fig. 3: Comparison of the interaction between *E. coli* FtsI anchor domain and FtsW from the EcFtsQLBWI model (left) and that between ttRodA and PBP2 (right, PDB 6PL5). Note the additional loop in PBP2 pointed by the arrow.

2. Truncation analysis that probes the role of the beta-sheet sandwich in the stabilization of the FtsQLBWI complex (Fig. 2) lacks key control experiments. For instance, the authors do not quantify expression levels of truncated constructs (e.g., via westerns) and do not include microscopy analysis showing cellular localization of truncated FtsI variants. In the absence of these data, it is unclear whether division activation/defects arise from differences in the cellular expression or localization of different variants or changes in binding/activity. It is also hard to interpret the seemingly self-contradictory results that truncation of the beta-sheet region for FtsL results in mislocalization of FtsL and cell division defects, while FtsI beta-sheet truncations promote division.

We have now replicated and quantified FtsI expression by Western blot and shown that the expression level of FtsI was reduced in the truncation with the most extreme cell-division defect (**Supplemental Fig. S14C**). Therefore, the defect we observed is associated with reduced FtsI expression levels, but it is not trivial to determine whether this arises from an increase in intrinsic FtsI stability, increased FtsI degradation with reduced incorporation into FtsQLBWI, or if truncation impacts FtsWI activity and this in turn impacts FtsI expression. We have left this as a question for future work. We also quantified immunofluorescence images of FtsI truncation cells and showed these mutants did not have a significant defect in midcell localization (**Supplemental Fig. S14E**).

Regarding the role of the β -sheet in the stabilization of the FstQLBWI complex, we note that addition of FtsL to the β -sheet was also observed experimentally in the *P. aeruginosa* cryo-EM structure. The FtsI extension of the β -sheet was not observed in the *P. aeruginosa* cryo-EM model, and the lack of conservation of C-terminal hydrophobic residues suggests that this is also a point of divergence (**Response Fig. 4**):

Pa RLMNVPPDNLPTATEQQQV-NAAPAKGGRG
 Ec RTMNI EPDALTTGDKNE **FVI**INQEGEGTGGRS
 * ** : ** * * . : : * * * . . . * * * .
 Δ14 Δ11 Δ10

Response Fig. 4 Alignment of *E. coli* and *P. aeruginosa* FtsI C-termini differ in hydrophobic residues; *E. coli* FtsI F576 and I578 (highlighted) interact with FtsQ, FtsL, and FtsB hydrophobic residues in the context of the β -sheet.

We expect that protein stability, localization, and protein-protein binding are coupled properties and that it will be difficult to measure their effects in isolation. Our goal in exploring FtsI truncations was to add to existing literature on FtsI C-terminal cleavage and rationally construct mutants in light of the predicted structure. Referring to **Response Fig. 4**, previous work identified cleavage to FtsI Δ 11, removing FtsI^{I578} which would destabilize FtsI C-terminal interaction with FtsQLB (Nagasawa, H. et al. *J. Bacteriol* 1989). In contrast, FtsI Δ 10 maintains all hydrophobic contacts and lowers the entropic cost of binding from constraining C-terminal residues, while FtsI Δ 14 eliminates all residues with β dihedral angles in predicted and simulated complexes. Together, our results suggest that reducing FtsI addition to the β sheet promotes division in a manner independent of FtsI stability. Additional experiments are required to elucidate the mechanism of this effect in greater detail. In the revision, we have clarified our discussion of FtsI truncation observations (lines 206 to 212).

In contrast to the subtle regulatory impact of the FtsI C-terminus, we expect that the FtsL strand is part of the conserved, core divisome structure (it is also predicted for all other divisome complexes we have looked at, including *B. subtilis* and *S. aureus*, **Supplemental Fig. S2** and observed experimentally for *P. aeruginosa* FtsQLBWI). We tested this hypothesis and found that FtsL truncations of increasing lengths led to increasing loss of conserved interactions with FtsB and FtsQ and produced increasingly severe defects in cell division (failure to complement FtsL depletion, increased cell length, and decrease in fraction of FtsL at midcell). Given the experimental confirmation of this interface by cryo-EM for the *P. aeruginosa* complex, we are satisfied that our observations are consistent with disrupting a conserved protein-protein interface. The interface could impact FtsQLBWI complex formation, activity of FtsQLBWI once complexes are formed, and stabilities of all complex components.

In the revision, we have added quantification of integrated fluorescence of mVenus-FtsL to our quantification of midcell localization (**Supplemental Fig. S13**).

We hope that these additional experiments explain the apparently self-contradictory of our results. In revision, we have carefully clarified our discussion of these points and quantified FtsI expression levels as described above.

3. For some mechanistic claims derived from AF models, there are no *in vitro* or *in vivo* experiments to complement MD simulations and to help validate the proposed mechanistic claims (Fig. 3-5). In general, it is hard to interpret MD simulations in the absence of mutational analysis coupled with either binding assays or co-localization imaging that can probe proposed changes in binding affinity directly. The MD simulation analysis itself is fairly limited, and it is not clear how some of the observed conformational changes contribute to binding/activity (as outlined in examples below).

Fig. 3: To what extent does a 1 Å shift in the packing between FtsL and FtsI at the AWI interface contribute to the overall binding energy?

The computational experiments reported in **Figs. 3–5** aim to investigate the molecular dynamics of interfaces identified in predicted structures as well as allosteric large-scale conformational change. We agree that mechanistic claims hinging only on MD would be problematic. For this reason, we focused on describing dynamics associated with (1) addition of FtsN, taking care to verify confidence in its predicted binding interface, (2) superfission mutations that rescue FtsN depletion, and (3) a dominant negative mutation that could be rescued by FtsN overexpression.

There are many observations that we made and did not include in the manuscript because we lack exactly this type of certainty from complimentary *in vivo* or *in vitro* data or a model including substrate for either FtsW or FtsI. The results in **Figs. 3–5** do suggest many testable hypotheses.

In the revision, we clarified descriptions of mechanisms and were careful to address limitations wherever we make mechanistic suggestions lacking *in vivo* or *in vitro* experimental data (lines 528 to 540).

In writing and editing the manuscript, we tried to avoid claims regarding binding affinity, but we agree that some of these appear in the manuscript (e.g. “further strengthening C-terminal interactions between FtsI and FtsQLB” pg 8 line 194). Additionally, some descriptions of changes observed are unnecessarily qualitative (e.g. “pack closely with hydrophobic residues” pg 9 line 251). MD does not directly provide binding free energies for complexes of this size, and therefore we cannot estimate how much the packing in the shift between FtsI and FtsL contribute to the overall binding energy.

Rather than quantify binding energy, in the revision we quantified solvent accessible surface area for hydrophobic FtsL residues in order to quantify increased interaction at the FtsL-FtsI interface going from FtsQLBWI to FtsQLBWI^{R167S} to FtsQLBWIN (lines 265 to 267 and 360 to 362). We also note that impacts of conformational change at this interface are observed in optimal-path analysis, where FtsI^{R167S} and especially the addition of FtsN increase the density of optimal paths running from FtsL to FtsW through FtsI (**Fig. 5E**).

In the revision, we have also clarified our language with respect to all apparent claims of changes in binding affinity to reflect that this is something which our simulations cannot directly measure.

Fig. 4: How much do local changes in the conformation of the FtsI anchor loop change substrate accessibility in the cavity of FtsW?

Our model does not contain a substrate and it is difficult for us to assess substrate accessibility. However, based on this suggestion we took another look at potential substrate cavities/channels in FtsW and observed that FtsW^{Y379} exhibits a conformational change during MD for FtsWI and FtsQL^{R61E}BWI simulations that may impact substrate transfer in complexes lacking key interactions with FtsL. This point requires further investigations.

Minor comments:

1. *The figures are hard to navigate: the authors could consider simplifying the cartoons and annotating them more explicitly, as well as using distinct colors for proteins that form close interfaces.*

In the revision we have added color labels to every figure for clarity and expanded the figure legends to ease navigation of the figures in a way that maintains essential detail and remains colorblind friendly.

Reviewer #1 (Remarks to the Author):

I am pleased to note that the authors have now included a comparison to the cryo-EM structure of the P. aeruginosa FtsQLBWI. This addition enhances the confidence in the results and interpretation of the coordinates obtained at the end of the MD simulations with the AF2 models. The authors have addressed my previous concerns and adequately incorporated the new data into their analysis. Based on this, I recommend the publication of this manuscript in Nat. communications.

Reviewer #2 (Remarks to the Author):

Based on the authors answers, I still do not fully believe in their modelling and think it is still too speculative to be published in Nature Communications. Maybe a more focused journal focusing on modelling membrane protein would be a better place ?

Reviewer #3 (Remarks to the Author):

The authors have clearly tried to address the concerns raised in the first review. The new controls they have included alleviate some of the concerns regarding the effect of mutations on protein expression and localization. However, since there is no data that directly probe binding or activity the authors are limited in the ability to draw clear conclusions about the role of individual interfaces. Further acknowledging the experimental limitations of this study should be done prior to publication.

Point-to-point Responses to Reviewers comments for Nature Communications manuscript NCOMMS-22-53130B: *Conformational changes in the essential E. coli septal cell wall synthesis complex suggest an activation mechanism*

For clarity, *we italicized and colored our responses in blue.*

REVIEWERS' COMMENTS

Reviewer #1 (Remarks to the Author):

I am pleased to note that the authors have now included a comparison to the cryo-EM structure of the P. aeruginosa FtsQLBWI. This addition enhances the confidence in the results and interpretation of the coordinates obtained at the end of the MD simulations with the AF2 models. The authors have addressed my previous concerns and adequately incorporated the new data into their analysis. Based on this, I recommend the publication of this manuscript in Nat. communications.

We thank the reviewer for their comments.

Reviewer #2 (Remarks to the Author):

Based on the authors answers, I still do not fully believe in their modelling and think it is still too speculative to be published in Nature Communications. Maybe a more focused journal focusing on modelling membrane protein would be a better place?

We thank the reviewer for their comments and note that we have revised the discussion section commenting on limitations of our study.

Reviewer #3 (Remarks to the Author):

The authors have clearly tried to address the concerns raised in the first review. The new controls they have included alleviate some of the concerns regarding the effect of mutations on protein expression and localization. However, since there is no data that directly probe binding or activity the authors are limited in the ability to draw clear conclusions about the role of individual interfaces. Further acknowledging the experimental limitations of this study should be done prior to publication.

We thank the reviewer for their comments. We agree that additional experiments perturbing divisive interfaces will add clarity to our understanding of divisive regulation. We have revised the limitations section of our discussion in light of these comments.